# Mechanisms of gene regulation by SRCAP and H2A.Z

Armelle Tollenaere [1] ✉, Enes Ugur[2,3,5], Susanna Dalla Longa[4,5], Cédric Deluz[1], Devin Assenheimer[4], J. Christof M. Gebhardt [4], Heinrich Leonhardt [3] & David M. Suter [1] ✉

Discriminating regulatory functions of chromatin composition from those of chromatin-modifying complexes is a central problem in gene regulation. This question remains unexplored in the context of histone variants and their dedicated chromatin remodelers. Here we dissect the distinct and cell cycle-dependent functions of Snf2 Related CREBBP Activator Protein (SRCAP) and H2A.Z in gene regulation of pluripotent stem cells. Using acute degradation of endogenous SRCAP, we uncover dynamic changes of H2A.Z occupancy and continuous requirement of SRCAP over the cell cycle. We also engineered an SRCAP mutant, defective for H2A.Z deposition, allowing us to distinguish H2A.Z-dependent and independent functions of SRCAP. We discover that SRCAP exhibits essential H2A.Z-independent functions in inhibiting DNA binding of dozens of pioneer transcription factors at enhancers by steric hindrance. In contrast, H2A.Z acts mainly as a transcriptional repressor gate-keeping the expression of lineage-specific genes. Our study establishes the catalytic-independent role of a chromatin remodeler in broadly regulating transcription factor binding, and demonstrates how a chromatin remodeler-histone variant pair orchestrates transcription to maintain self-renewal and plasticity of pluripotent stem cells.

In Eukaryotes, most DNA is wrapped around nucleosomes made of histone octamers. While two copies of the four canonical histones make up most nucleosomes, histone variants can replace them in specific positions of the genome. H2A.Z is a highly conserved H2A histone variant essential for zygotic genome activation in *Drosophila* and zebrafish and for post-implantation development in the mouse[1–3]. In vertebrates, there are two different isoforms of H2A.Z (H2A.Z.1 and H2A.Z.2) that differ by only three amino acids[4]. H2A.Z plays a central role in self-renewal and lineage commitment of multiple types of stem cells, including mouse pluripotent stem cells (PSCs)[5–7]. Both Snf2-related CREBBP activator protein (SRCAP) and Tip60/p400 remodeler

complexes ensure H2A.Z deposition in mammals[8,9]. In mouse PSCs, H2A.Z deposition is mainly achieved by the SRCAP complex, made of the SRCAP protein bearing the catalytic ATPase activity and 9 other subunits[10]. The SRCAP complex binds to canonical nucleosomes and swaps the H2A-H2B dimer with an H2A.Z-H2B dimer[8]. It was also reported to facilitate the recruitment of some transcriptional regulators, but whether this function is independent of its role in depositing H2A.Z is unclear[11–13].

In PSCs, H2A.Z is broadly enriched at active and bivalent/poised promoters and enhancers[14,15]. H2A.Z has been proposed to regulate transcription at multiple levels, i.e., by modulating chromatin

[1]Ecole Polytechnique Fédérale de Lausanne, School of Life Sciences, Institute of Bioengineering, Lausanne, Switzerland. [2]Department of Proteomics and Signal Transduction, Max-Planck Institute of Biochemistry, Martinsried, Germany. [3]Faculty of Biology and Center for Molecular Biosystems (BioSysM), Human Biology and BioImaging, Ludwig-Maximilians-Universität München, Munich, Germany. [4]Institute of Experimental Physics and IQST, Ulm University, Albert-Einstein-Allee 11, Ulm, Germany. [5]These authors contributed equally: Enes Ugur, Susanna Dalla Longa. ✉e-mail: armelle.tollenaere@epfl.ch; david.suter@epfl.ch

accessibility, repressing DNA methylation, altering transcription factor (TF) binding and enhancing RNA polymerase II pausing[16–20]. H2A.Z nucleosomes protect shorter DNA segments compared to canonical nucleosomes, which was suggested to reflect nucleosome destabilization and to facilitate protein-DNA interactions[21].

Mechanistic understanding of transcriptional regulation by SRCAP and H2A.Z remains obscured by conflicting evidence and technical challenges. First, H2A.Z was reported to either increase or decrease both chromatin accessibility and TF binding[5,16,18,22,23]. Second, whether H2A.Z-containing nucleosomes are evicted or replaced with canonical nucleosomes when H2A.Z is depleted is unclear[5,24,25]. Thus, the specific function of H2A.Z with respect to H2A has not yet been addressed in vivo. Third, the proposed role of H2A.Z in inhibiting DNA methylation in mammals is only supported by weak evidence that is either correlative or uses slow H2A.Z knockdown approaches may cause indirect effects[26,27]. Fourth, depletion of H2A.Z causes genome-wide redistribution of the yeast SRCAP homolog (SWR1)[28], and nucleosomal H2A.Z inhibits human SRCAP binding[23], potentially leading to SRCAP-mediated changes in recruiting transcriptional regulators[29].

Here, we use the rapid degradation of endogenous SRCAP to dissect the direct and acute consequences of SRCAP loss and analyze the function of H2A.Z with respect to the canonical-H2A histone. Overexpressing a SRCAP mutant defective for H2A.Z deposition allowed us to uncouple H2A.Z deposition from other SRCAP functions in transcriptional regulation. Our study reveals how SRCAP and H2A.Z coordinate their distinct functions to orchestrate gene regulation governing self-renewal and plasticity.

## Results

### Dynamic and cell cycle-dependent SRCAP-mediated H2A.Z positioning

To study the dynamics of H2A/H2A.Z exchange by SRCAP, we generated a CGR8 mouse embryonic stem cell (mESC) line allowing rapid degradation of the catalytic subunit of the SRCAP complex (SRCAP), using the auxin-inducible protein degradation system[30]. We knocked-in YPet-AID (YA) into the 3′end of both alleles of *srcap* and stably expressed the OsTir1 receptor to generate the SRCAP-Ypet-AID Tir1 (SYAT) cell line. We validated the pluripotency of SYAT cells by Alkaline Phosphatase assay and RT-QPCR of pluripotency TFs (Supplementary Fig. S1a), nuclear interphase localization and previously described mitotic chromosome association of SRCAP[31] (Supplementary Fig. S1b). CUT&RUN and CUT&TAG revealed SRCAP-YA enrichment at promoters and preferential colocalization with active transcription start sites (TSS) and H2A.Z peaks (Supplementary Fig. S1c, d and Fig. 1a), as reported for SWR1 in yeast[28]. Additionally, by re-analyzing published data[32], we confirmed that SRCAP-YA was enriched at the same genomic locations as endogenous SRCAP in wt mESCs (Supplementary Fig. S1e). Upon addition of indole-3-acetic acid (IAA), SRCAP-YA was rapidly depleted (Fig. 1b and Supplementary Fig. S1f), and H2A.Z signal on mitotic chromosomes was decreased (Supplementary Fig. S1g).

We then performed H2A.Z and H2A ChIP-seq (see Methods) at 2, 4, 6, or 8 h after IAA addition. H2A.Z was depleted and replaced by H2A (Fig. 1c, d) at 98% of H2A.Z ChIP-seq peaks, with the majority of peaks (72%) losing most H2A.Z within 2 h (Supplementary Fig. S1f). H2A.Z turnover at regions specifically enriched for H2A.Z.1 or H2A.Z.2 (using data from ref. 20) was similar, suggesting a comparable impact of SRCAP depletion on H2A.Z.1 and H2A.Z.2 (Supplementary Fig. S1h). The rate of SRCAP-mediated H2A.Z exchange was highest at active promoters, followed by active enhancers, and slightly slower at bivalent and poised enhancers (Fig. 1e, f). The pace of H2A.Z loss was particularly fast at nucleosomes surrounding TSS of active genes (Fig. 1g) and scaled with their transcriptional activity (Supplementary Fig. S1i). At bivalent promoters, while H2A.Z occupancy was high,

SRCAP-mediated H2A.Z turnover was slower. This is in line with slower eviction of H2A.Z leading to a less pronounced recruitment of SRCAP at these loci (Fig. 1a). Also note that we cannot exclude that Ep400-Tip60 also contributes to H2A.Z deposition at bivalent TSS. Loci with the highest H2A.Z turnover were initially highly enriched for SRCAP, the SWI/SNF, NuRD and Ep400 chromatin remodelers, and several PSC TFs (Fig. 1h and Supplementary Fig. S1j), suggesting their role in H2A.Z turnover regulation. We next examined H2A.Z colocalization with the large number of regions depending on the pioneer TFs (pTFs) OCT4 and SOX2 for chromatin accessibility maintenance in PSCs[33,34]. H2A.Z was rapidly lost in these regions upon SRCAP depletion (Supplementary Fig. S1k). Furthermore, knockdown of OCT4 (see Methods and ref. 35) rescued H2A.Z occupancy at OCT4-occupied regions (Supplementary Fig. S1l), suggesting that OCT4 mediates H2A.Z eviction. These findings are in line with H2A.Z nucleosome eviction by the transcriptional machinery and gene regulatory factors[36,37].

As TF binding, chromatin remodeler activity and transcriptional activity vary across the cell cycle, we determined H2A.Z positioning in early G1 (EG1), late G1 (LG1), S, G2/M phases, or mitosis (Methods and refs. 34,38). Global variations of H2A.Z through the cell cycle were modest (Fig. 2a), and H2A.Z was mostly located at active and bivalent/poised regulatory elements (Supplementary Fig. S2a, b). H2A.Z was lost at active promoters and enhancers at the mitosis-EG1 transition, but accumulated at bivalent and poised enhancers (Fig. 2b). We neither observed a mitotic shift of H2A.Z on the TSS nor a mitotic H2A.Z depletion at promoters as reported in different cell types[39,40], suggesting cell type-specificity of H2A.Z positioning. We then classified regulatory elements according to changes in H2A.Z occupancy between the G2 phase and mitosis (Fig. 2c and Supplementary Fig. S2c). Global H2A.Z levels were equalized during mitosis at promoters and enhancers, in line with a quantitative resetting of the H2A.Z landscape before cells enter the next cell cycle (Fig. 2d and Supplementary Fig. S2d). Promoters in cluster 1 and cluster 3 showed enrichment for Gene Ontology (GO) terms associated with lineage specification and general cellular functions (Supplementary Fig. S2e), respectively. Highly-expressed genes were more enriched in cluster 3 (Fig. 2e), but enhancer activity levels were comparable in all enhancer clusters (Supplementary Fig. S2f). Next, we performed ChIP-seq on the acetylated form of H2A.Z (acH2A.Z), associated with active regulatory elements and positively correlated with transcription levels[41]. Mitotic acH2A.Z levels were decreased in all three promoter clusters (Fig. 2f and Supplementary Fig. S2g, h), reflecting a shift in the equilibrium between acetylation and deacetylation during mitosis.

We next investigated SRCAP-dependent H2A.Z positioning during mitosis. We treated cells with IAA for 1 h and sorted asynchronous and mitotic cells depleted for SRCAP. 1 h-IAA treatment decreased H2A.Z occupancy genome-wide (Fig. 2g and Supplementary Fig. S2i) at bivalent/poised promoters and enhancers in mitosis but not in interphase (Fig. 2h and Supplementary Fig. S2j). While we noted a small increase in H2A.Z enrichment at bivalent promoters in asynchronous cells upon 1 h IAA treatment, the biological meaning of this finding is unclear (Fig. 2h). In contrast, 1 h IAA treatment in both mitosis and interphase decreased H2A.Z levels at active promoters and enhancers, in line with their high H2A.Z turnover rate (Fig. 2h and Supplementary Fig. S2j). This suggests that SRCAP-mediated H2A.Z turnover is accelerated in mitosis despite very low transcriptional activity[42], indicating transcription-independent mitotic H2A.Z eviction.

### SRCAP depletion unleashes the expression of master regulators of cell fate

We next performed nuclear RNA-seq after IAA treatment for 2, 4, 6, or 8 h in SYAT cells or wt CGR8 cells for 4 h to account for unspecific effects of IAA. We quantified both intronic reads, which accurately reflect acute transcriptional activity[43], and exonic reads. In wt CGR8 cells, 4 h of IAA treatment did not impact transcription

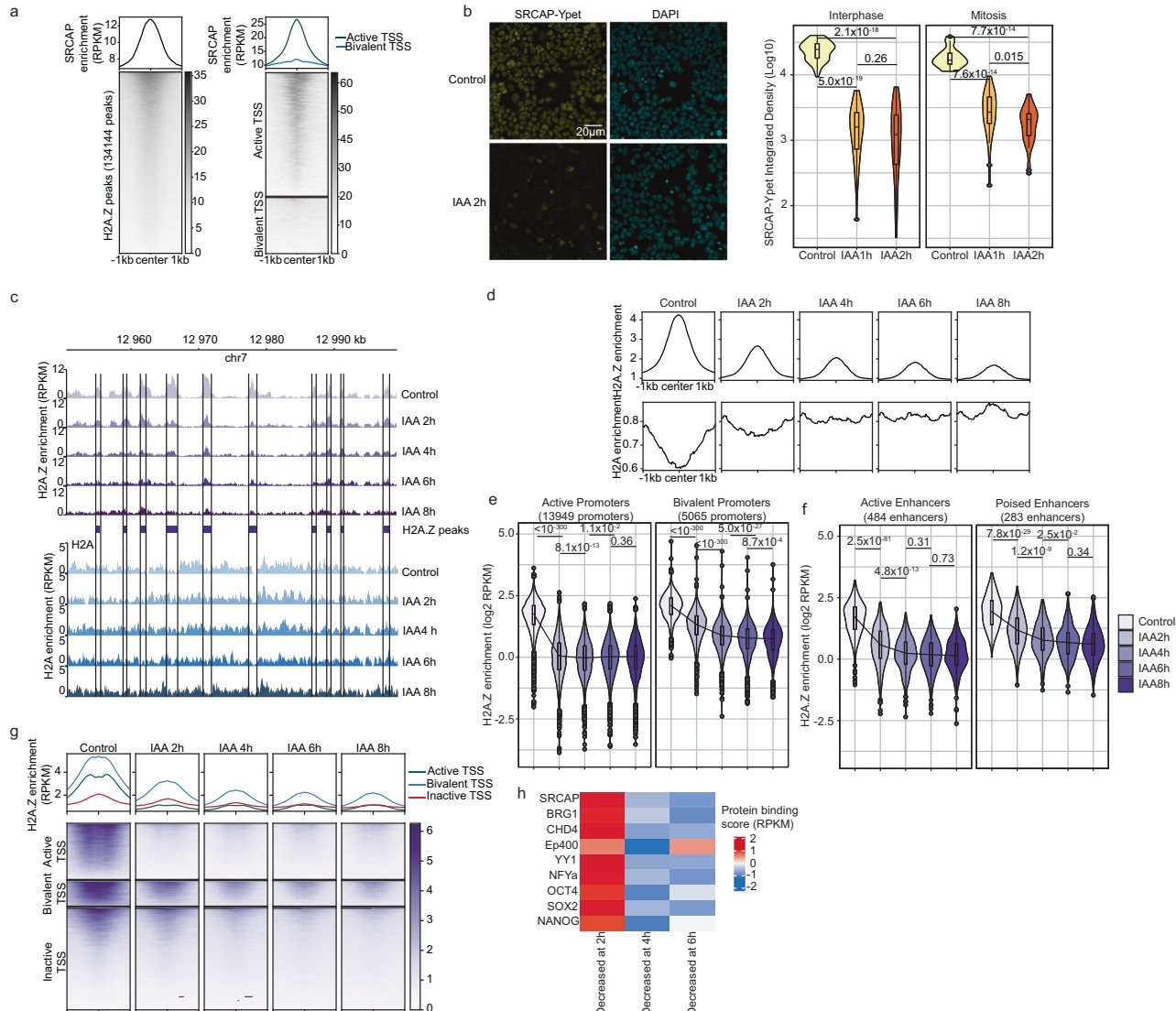

**Fig. 1 | SRCAP degradation leads to the rapid replacement of H2A.Z by H2A.**
**a** SRCAP-YPet-AID genomic mapping by CUT&RUN at H2A.Z peaks in control SYAT (left panel) and active (19,308) and bivalent (8763) TSS (right panel). **b** Left panel: Microscopy snapshots of SYAT cells, treated with IAA or control for 2 h and stained with DAPI. Right panel: Quantification of total SRCAP-YA signal in interphase and mitotic cells treated with IAA or control. $N = 50$ cells per condition. Boxes: interquartile range (IQR); whiskers: 1.5xIQR; dots: outlier points. **c** Genome tracks of H2A.Z and H2A ChIP-Seq in SYAT cells treated with IAA. **d** ChIP-Seq profiles of

H2A.Z and H2A at H2A.Z peaks (143561 peaks) in SYAT cells treated with IAA. **e, f** H2A.Z enrichment at active and bivalent promoters (**e**) and enhancers (**f**) upon SRCAP degradation. Black line: changes in median values. Boxes: interquartile range (IQR); whiskers: 1.5xIQR; dots: outlier points. **g** ChIP-seq metaplots and heatmaps of H2A.Z enrichment around the TSS of active (19308), bivalent (8763), and inactive (35247) genes. **h** Z-score enrichment of different DNA-binding proteins for the different H2A.Z turnover groups. $p$ values were determined by a two-sided pairwise Wilcoxon test and are indicated on the graphs.

(Supplementary Fig. S3a). In SYAT cells, changes in intronic reads were already apparent after 2 h and peaked after 6 h of IAA treatment, with 211 and 276 genes up- and downregulated, respectively (Fig. 3a and Supplementary Data 1). Mature transcripts were more gradually affected starting from 2 h of IAA treatment (Supplementary Fig. S3b). These results contrast with H2A.Z knockdown in mESCs, which mainly results in upregulated genes[20,44] that poorly overlap with the genes we identified (Supplementary Fig. S3c). The pace of H2A.Z removal was comparable between transcriptionally-affected and unaffected genes (Supplementary Fig. S3d). Bivalent genes were markedly over-represented among upregulated genes upon SRCAP depletion (Fig. 3b), but no functional gene category was overrepresented in down-regulated genes (Supplementary Fig. S3e). Upregulated genes were highly enriched in DNA-binding proteins, notably in lineage-specific transcription factors (Fig. 3c and Supplementary Fig. S3f). Among those were TFs associated with differentiation towards germ layers,

especially mesoderm (Gata3, Gata6, BMP4, Meis1, Lefty2), or with pluripotency maintenance (Nanog, Myc, Trim28, and Zfp42) (Supplementary Fig. S3g and Supplementary Data 1). In contrast, down-regulated genes were enriched in ubiquitously expressed genes (Fig. 3d). We next quantified the number of promoter-enhancer (P-E) loops formed for each gene as well as the P-E loop strength (Methods). Upregulated genes formed more promoter-enhancer loops as well as slightly stronger loops than downregulated genes (Fig. 3e and Supplementary Fig. S3h), in line with enhancer-dependent regulation of lineage-specific genes[45]. We then quantified changes in the proteome of the 4 h IAA-treated cells by Mass Spectrometry (Methods). Only 52 and 30 proteins were significantly increased and decreased in their levels, respectively (Fig. 3f and Supplementary Data 2). GO terms associated with DNA binding and transcriptional regulation were enriched in upregulated proteins (Supplementary Fig. S3i), in line with

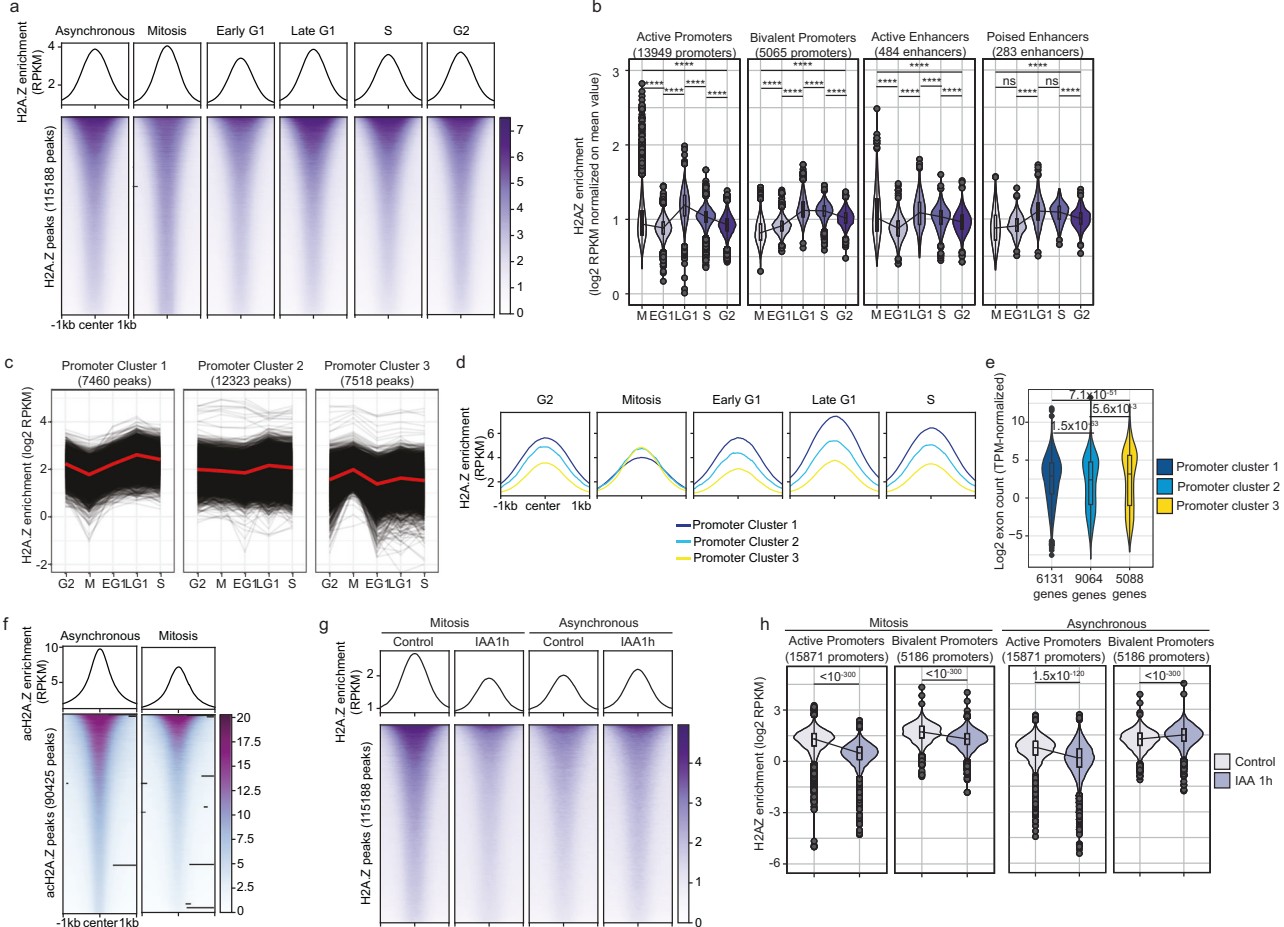

**Fig. 2 | H2A.Z deposition across the cell cycle. a** Heatmaps and metaplots of H2A.Z enrichment centered on the summit of H2A.Z peaks across the cell cycle. **b** Mean-normalized H2A.Z enrichment calculated from H2A.Z-scores across the cell cycle in promoters or enhancers containing H2A.Z peaks, classified according to their activity. Black line: changes in median values. Boxes: interquartile range (IQR); whiskers: 1.5xIQR; dots: outlier points. *p* values are indicated in Supplementary Data 5. **c** H2A.Z enrichment in promoter peaks in regions where mitotic H2A.Z was decreased, stable or increased with respect to G2 and EG1 phases. **d** H2A.Z enrichment around H2A.Z peaks in the clusters shown in (**c**). **e** Exon counts of genes regulated by each of the three promoter clusters. RNA-Seq data from SYAT control samples, performed in two biological replicates were used. Boxes: interquartile range (IQR); whiskers: 1.5xIQR; dots: outlier points. **f** Heatmaps and metaplots of acH2A.Z enrichment centered on the summit of acH2A.Z peaks in asynchronous versus mitotic cells. **g** Heatmap of H2A.Z enrichment centered on the summit of H2A.Z peaks in mitotic or asynchronous SYAT cells treated for 1 h with IAA. **h** H2A.Z enrichment in promoters of mitotic or asynchronous SYAT cells treated for 1 h with IAA. Black line: changes in median values. Boxes: interquartile range (IQR); whiskers: 1.5xIQR; dots: outlier points. *p* values were determined by a two-sided pairwise Wilcoxon test, ****p* < 0.0001, ns nonsignificant.

upregulated mRNAs. Note that we expect protein levels to reflect changes that we observed at the mRNA levels only at longer timescales.

We next investigated the impact of mitotic SRCAP depletion on gene reactivation during EG1 (Methods). We treated cells with IAA for 1 h, followed by sorting of EG1 cells or LG1 cells (undergoing SRCAP depletion during mitosis and G1, respectively). Mitotic SRCAP depletion resulted mainly in downregulation of genes in EG1 (Fig. 3g), many of which were housekeeping genes (Supplementary Fig. S3j). In contrast, SRCAP depletion during G1 resulted in a smaller number of downregulated genes (Fig. 3h). This suggests that SRCAP activity during mitosis and at mitotic exit is mainly required for transcriptional reactivation in early G1.

## SRCAP depletion alters histone modifications and nucleosome properties

To understand the mechanisms underlying changes in transcriptional activity upon SRCAP depletion, we analyzed changes in histone post-translational modifications by ChIP-seq and nucleosome properties by MNase-seq. At 4 h of IAA, H3K4me3 levels were down in 66.6% of H3K4me3-enriched regions (Fig. 4a and Supplementary Fig. S4a) and

66.8% of H3K4me3 peaks also enriched for H2A.Z (Supplementary Fig. S4b), with a marked decrease at bivalent promoters (Fig. 4b). This is in line with the role of H2A.Z in recruiting SET1/COMPASS complexes[5], which allows H3K4me3 maintenance[46]. Changes in H3K4me3 enrichment at +1 nucleosomes were mildly correlated with transcriptional changes for both active and bivalent genes (Fig. 4c and Supplementary Fig. S4c), and promoters of downregulated genes displayed a more pronounced decrease of H3K4me3 (Supplementary Fig. S4d), in line with the role of H3K4me3 in fostering transcriptional activity[47,48].

H3K27me3 levels decreased at 27.6% of regions (4799 peaks) and increased at 19.6% of regions (3402 peaks), including at bivalent promoters (Fig. 4a, d and Supplementary Fig. S4e). H3K27me3 peaks also enriched for H2A.Z presented the same trend (Supplementary Fig. S4f). H2AK119ub1 displayed a massive increase at most enriched loci (52.6%, 20881 peaks), especially at bivalent promoters (Fig. 4a, e and Supplementary Fig. S4g). This trend was even more clear at H2AK119ub1, and H2A.Z shared peaks (Supplementary Fig. S4h). Additionally, changes in H3K27me3 and H2AK119ub1 levels do not explain transcriptional changes (Supplementary Fig. S4i, j).

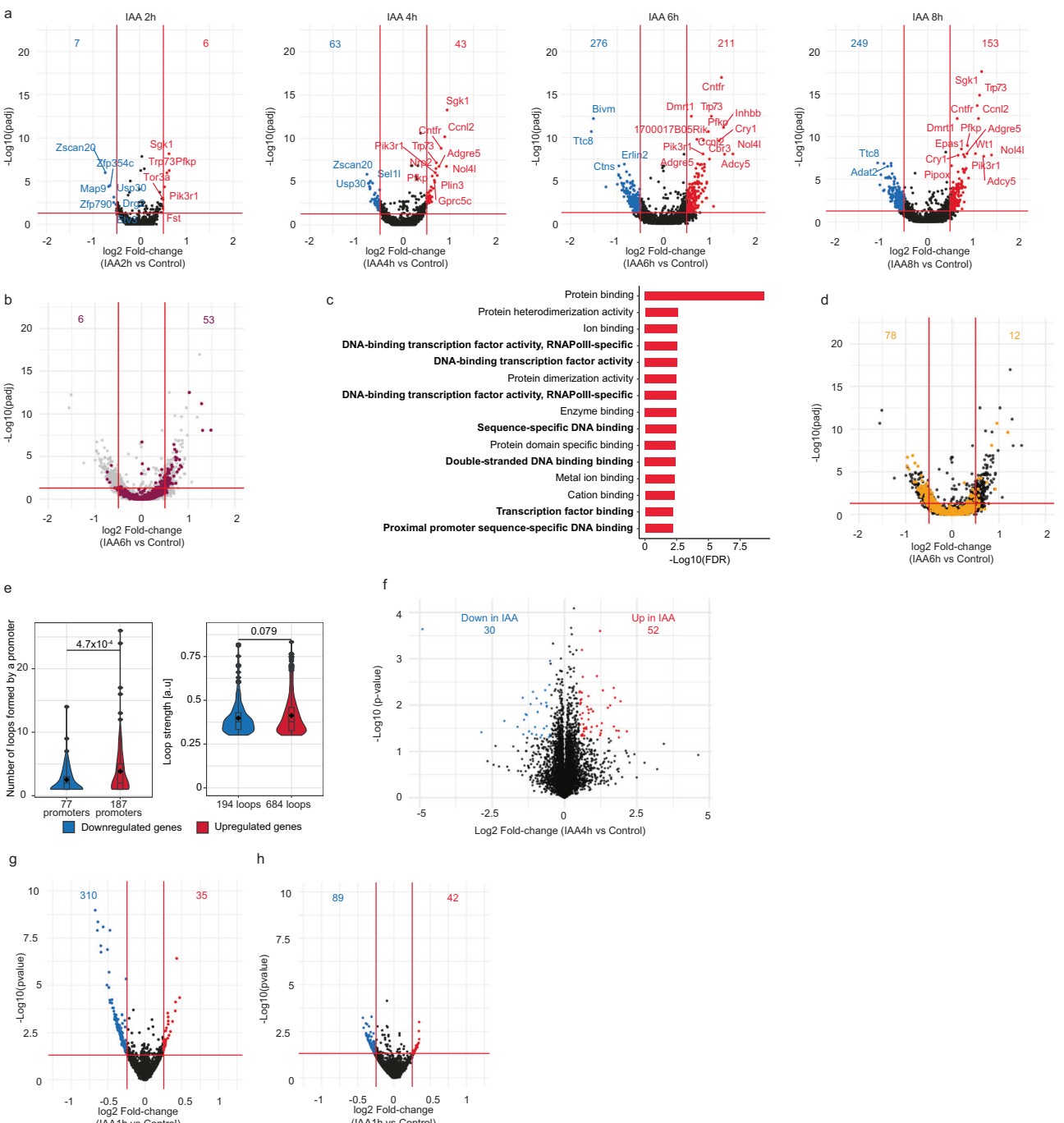

**Fig. 3 | SRCAP degradation leads to transcriptional deregulation. a** Fold-change of intronic reads and Benjamini–Hochberg (BH) adjusted $p$ value (computed from Wald test performed by DESeq2) for SYAT cells treated for 2, 4, 6, and 8 h with IAA. Numbers: genes significantly downregulated (blue) or upregulated (red) with a threshold of |log2FC| >0.5 and adjusted $p$ value <0.05. **b** Fold-change of intronic reads and BH-adjusted $p$ value (DESeq2 default Wald test) for SYAT cells treated for 6 h with IAA. Genes regulated by a bivalent promoter are colored. **c** Top 15 GO terms enriched for genes upregulated at 6 h with the category "molecular function". **d** Fold-change of intronic reads and adjusted $p$ value for SYAT cells treated for 6 h with IAA. Ubiquitously-expressed genes are colored. **e** Number (left panel) and strength (right panel) of promoter-enhancer loops formed for genes found to be

downregulated or upregulated at any time point. Data from ref. 113. $p$ values were determined by a two-sided Welch two-sample $t$-test, and are indicated on the plots. **f** Changes in proteome composition in SYAT cells treated with IAA for 4 h compared to control. Numbers: proteins enriched (blue) or depleted (red) from the proteome upon IAA treatment with a threshold of |log2FC| >0.5 and $p$ value <0.05. $p$ values were determined using a two-sided Student's $t$-test. **g, h** Fold-change of intronic reads, for SYAT cells treated for 1 h with IAA and sorted in EG1 (**g**) or LG1 (**h**). Numbers: genes significantly downregulated (blue) or upregulated (red) with a threshold of |log2FC| >0.25 and $p$ value <0.05. $p$ values were determined using the DESeq2 default Wald test.

We next performed MNase-seq using high or low concentrations of MNase (Methods) on SYAT cells treated with control or IAA for 4 h. We classified nucleosomes in four classes according to initial H2A.Z enrichment scores. H2A.Z enrichment scaled with nucleosome

occupancy at regulatory elements (Fig. 4f and Supplementary Fig. S4k). However, this coupling was also observed upon SRCAP/H2A.Z depletion, suggesting that nucleosome occupancy is not primarily regulated by H2A.Z but is mostly dependent on the genomic context.

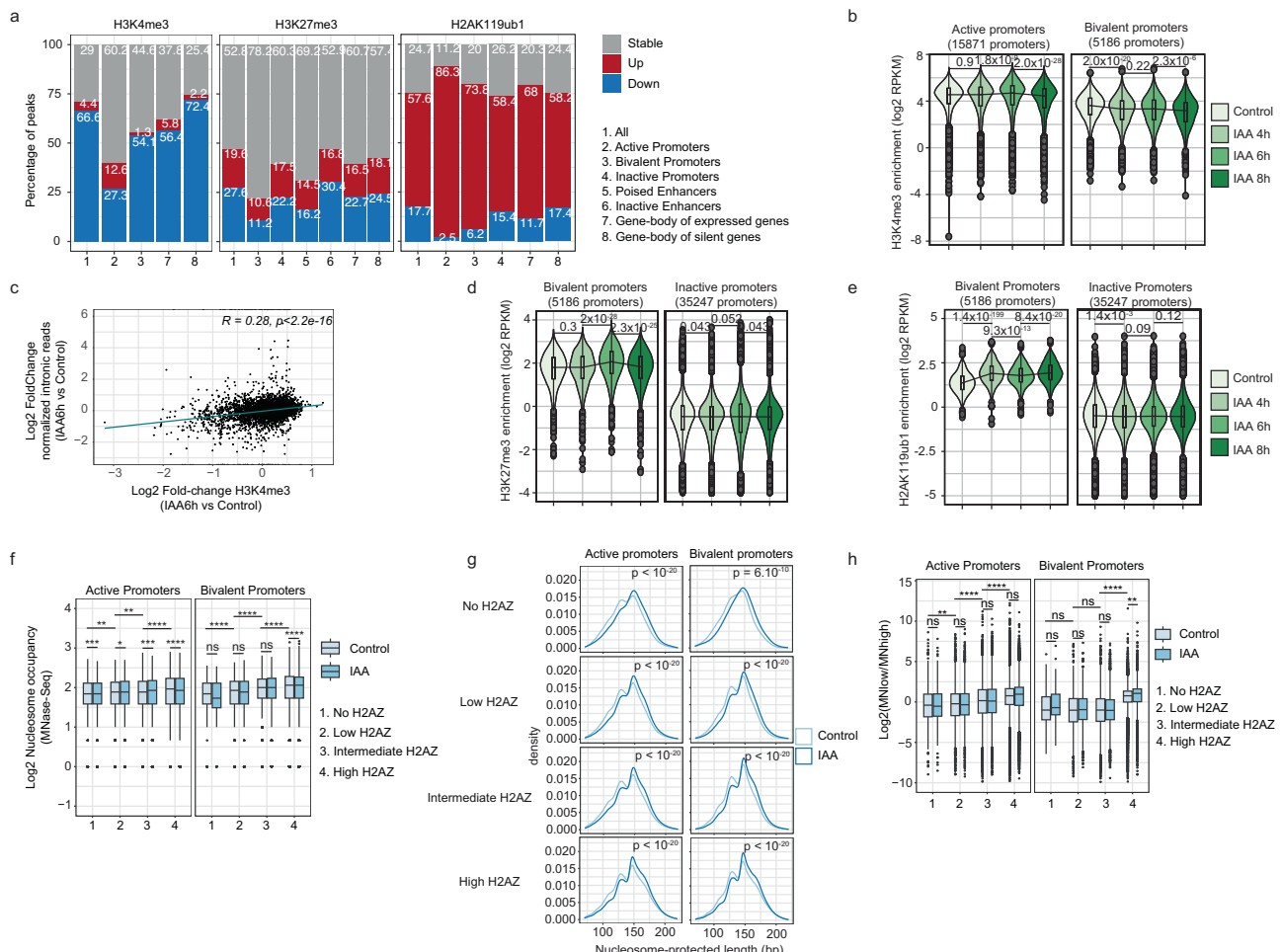

**Fig. 4 | Impact of SRCAP/H2A.Z loss on nucleosome properties and post-translational modifications. a** Percentage of different types of genomic regions with increased, stable or decreased H3K4me3, H3K27me3 and H2AK119ub1 marks. **b** H3K4me3 enrichment at active and bivalent promoters, in SYAT cells treated with control or IAA for 4, 6, and 8 h. Black line: changes in median values. Boxes: interquartile range (IQR); whiskers: 1.5xIQR; dots: outlier points. **c** Scatterplot of intronic reads of genes controlled by active promoters as a function of H3K4me3 enrichment changes between 6 h of IAA treatment and control. Upper right corner: Pearson correlation coefficient. **d, e** H3K27me3 (**d**) and H2AK119ub1 (**e**) enrichment at bivalent and inactive promoters in SYAT cells treated with control or IAA for 4, 6, and 8 h. Black line: changes in median values. Boxes: interquartile range (IQR); whiskers: 1.5xIQR; dots: outlier points. **f** Nucleosome occupancy at promoters in SYAT cells as a function of H2A.Z levels, in control or after 4 h of IAA treatment. Boxes: interquartile range (IQR); whiskers: 1.5xIQR; dots: outlier points. Exact *p*

values are provided in Supplementary Data 5. Numbers of nucleosomes analyzed are indicated in supplementary Data 4. **g** MNase-Seq fragment densities in SYAT cells treated with control or IAA for 4 h, as a function of H2A.Z enrichment and the class of promoter where nucleosomes are localized. **h** Fragile nucleosomes in nucleosomes of different types of promoters, as a function of H2A.Z levels. Boxes: interquartile range (IQR); whiskers: 1.5xIQR; dots: outlier points. Exact *p* values are provided in supplementary Data 5. Numbers of nucleosomes analyzed are indicated in supplementary Data 4. *p* values comparing H2A.Z levels in boxplots were determined by a pairwise two-sided Wilcoxon test. *p* values comparing CTRL and IAA conditions in the boxplot were determined by a two-sided Welch two-sample *t*-test. *p* values comparing fragment length distributions were determined by a Kolmogorov–Smirnov test. ****$p < 0.0001$; ***$p < 0.001$; **$p < 0.01$; *$p < 0.05$, ns nonsignificant.

The size of nucleosome-protected fragments increased upon SRCAP degradation independently of initial H2A.Z levels in regions containing these nucleosomes (Fig. 4g and Supplementary Fig. S4l). While agreeing with the reported unwrapped state of H2A.Z nucleosomes[21], this also raises the possibility that SRCAP unwraps nucleosomes independently of H2A.Z deposition, similarly to SWR1 in vitro[49,50]. We then focused on fragile nucleosomes, defined by a high signal ratio between MN-low versus MN-high digestion and enriched around TSS and TF binding sites[51,52]. Nucleosome fragility scaled with H2A.Z levels at active and bivalent regulatory regions (Fig. 4h and Supplementary Fig. S4m), in agreement with reported H2A.Z enrichment in fragile nucleosomes[53–55]. Surprisingly, IAA did not induce a global decrease of nucleosome fragility (Fig. 4h and Supplementary Fig. S4m). Furthermore, nucleosome fragility was unchanged at +1 nucleosomes of downregulated genes and increased at upregulated genes

(Supplementary Fig. S4n), suggesting that fragility depends on the genomic context but not on H2A.Z occupancy. Therefore, alteration of nucleosome properties does not explain the transcriptional changes mediated by SRCAP depletion.

## SRCAP depletion unleashes transcription factor binding
We next performed ATAC-seq after treating SYAT cells for 2, 4, 6, or 8 h with IAA. Overall chromatin accessibility was unchanged within the first 6 h of SRCAP degradation (Supplementary Fig. S5a). Promoters were mostly unaffected in their accessibility, while many distal inter-genic regions displayed increased or decreased accessibility (Fig. 5a, b). To understand the mechanisms underlying these changes, we analyzed TF footprints in accessible regions using the TOBIAS bioin-formatic pipeline[56]. In wt CGR8 cells, a 4 h IAA treatment had an anecdotal impact on TF footprinting (Supplementary Fig. S5b and

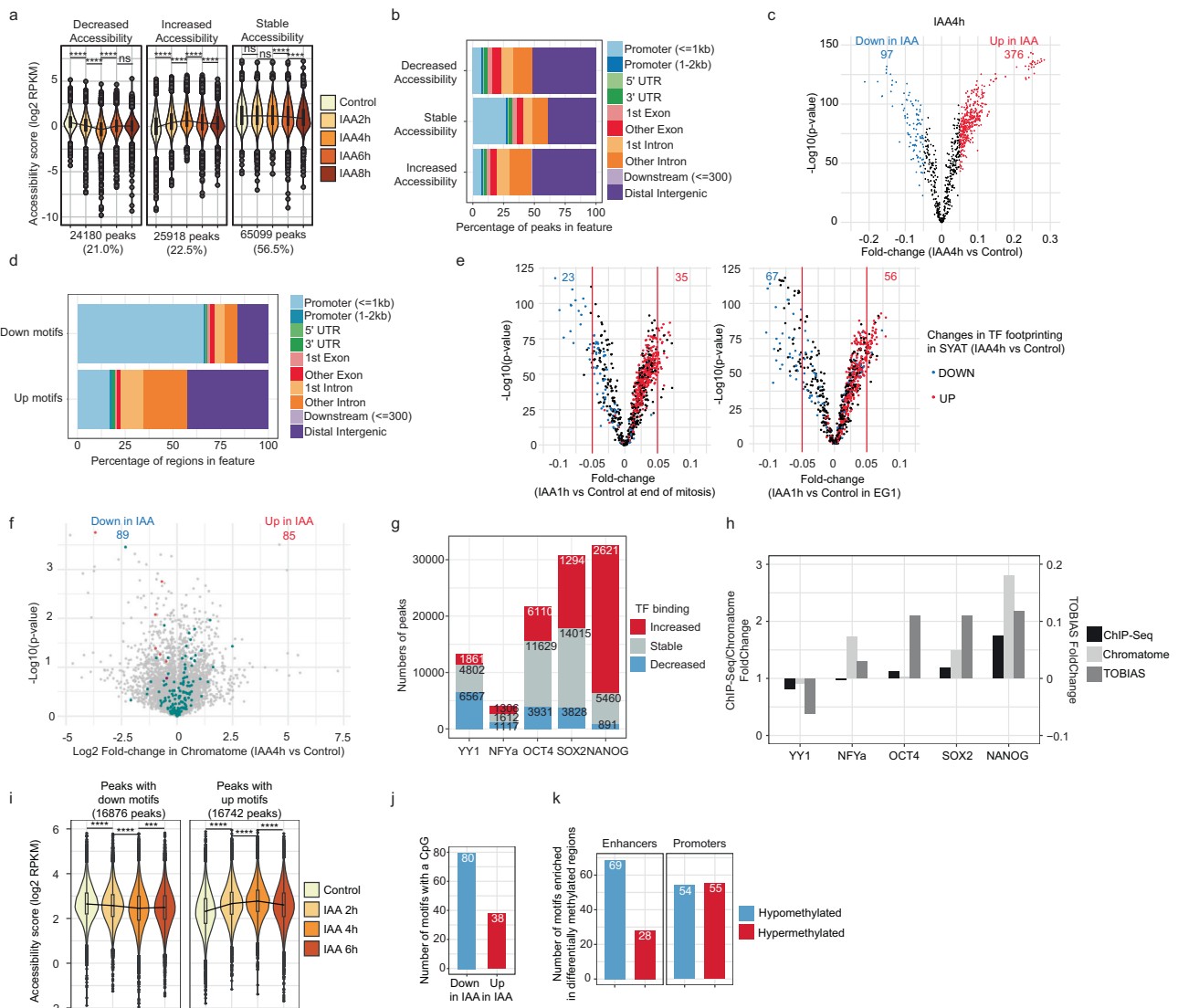

**Fig. 5 | SRCAP depletion alters transcription factor binding. a** Chromatin accessibility changes in SYAT cells treated with control or IAA for 2, 4, 6, and 8 h. Exact *p* values are provided in supplementary Data 5. Black line: changes in median values. Boxes: interquartile range (IQR); whiskers: 1.5xIQR; dots: outlier points. **b** Percentage of different types of genomic regions in accessible regions belonging to accessibility classes defined in (**a**). **c** Fold-change of TF footprinting calculated using TOBIAS in SYAT cells treated for 4 h with IAA versus control. Numbers: TF motifs losing (blue) or gaining (red) binding upon IAA treatment with |FC| >0.05 and *p* value <0.0001 (*p* value provided by TOBIAS). **d** Percentage of occurrence of all motifs, for TFs predicted to gain or lose binding upon IAA treatment, for different genomic regions. **e** Fold-change of TF footprinting calculated using TOBIAS in SYAT cells at the end of mitosis (left panel) or EG1 (right panel). Numbers: TF motifs predicted to lose (blue) or gain (red) binding upon IAA treatment with |FC| >0.05 and *p* value <0.0001 (*p* value provided by TOBIAS). Colors: Predicted TF footprinting changes in SYAT treated 4 h with IAA versus control. **f** Chromatome

analysis. Turquoise: TFs; red: members of SRCAP complex; purple: H2A.Z. Numbers: proteins enriched (blue) or depleted (red) from the chromatome upon IAA treatment with |log2FC| >0.5 and *p* value <0.05. *p* values were determined using a two-sided Student's *t*-test. **g** Numbers of peaks with an increased or decreased TF ChIP-seq enrichment after 4 h of IAA treatment in SYAT cells. **h** Fold-change in TF binding determined by ChIP-seq, chromatome, and TOBIAS prediction analysis. **i** Chromatin accessibility changes upon IAA treatment for TF motifs predicted to gain or lose binding. Exact *p* values are provided in Supplementary Data 5. **j** Number of motifs containing at least one CpG, in motifs predicted by TOBIAS to lose or gain binding upon SRCAP depletion. **k** Numbers of motifs enriched in H2A.Z peaks located in promoters or enhancers and displaying cytosine hyper- or hypomethylation upon SRCAP/H2A.Z loss. Unless indicated otherwise, *p* values were determined by a two-sided pairwise Wilcoxon test; ****p* < 0.0001; ****p* < 0.001; ns nonsignificant.

Supplementary Data 3). In SYAT cells, hundreds of TF motifs already displayed significant changes in footprinting after 2 h of IAA treatment, and these alterations were very consistent across time points (Fig. 5c and Supplementary Fig. S5c). About four times more motifs had their footprints enriched (up motifs) than depleted (down motifs) at all time points after IAA treatment (Fig. 5c and Supplementary Fig. S5c). Up and down motifs were mostly located in distal regulatory regions and promoter regions, respectively (Fig. 5d and Supplementary Fig. S5d). As promoter- and enhancer-regulated genes tend to be downregulated

and upregulated upon SRCAP/H2A.Z depletion, respectively (Fig. 3e), this suggests that changes in TF binding could contribute to transcriptional changes caused by SRCAP depletion. We next asked whether the predominant transcriptional downregulation after mitotic IAA treatment (Fig. 3g) could be caused by a shift towards more downregulated TF footprints. We performed 1 h of IAA or control treatment, and sorted cells enriched for the end of M-phase (see Methods) and in EG1 to perform ATAC-seq prior to and concurrently with EG1 transcriptional changes, respectively. Changes in TF footprinting were

more pronounced for down relative to up motifs as compared to longer IAA treatments in asynchronous cells (Fig. 5e), in line with transcriptional downregulation upon mitotic SRCAP/H2A.Z loss.

As pTFs can bind nucleosomal DNA, we reasoned that their binding might be particularly sensitive to nucleosome composition. We quantified the number of pTFs enriched in up vs downregulated motifs, using a classification of pTFs based on refs. 57,58 (see Methods). 87 out of 376 up-TFs (23.1%) but only 7 out of 97 (7.2%) down-TFs after 4 h of IAA treatment were known pTFs (Supplementary Fig. S5e). Therefore, pTF binding is generally impaired by the presence of SRCAP and/or H2A.Z.

To gain a broader view of changes in protein-DNA interactions induced by SRCAP depletion, we treated cells with IAA for 4 h and analyzed changes in proteins bound to chromatin (chromatome) (ref. 59 and Methods). We found 174 proteins significantly up- or down-regulated. In line with TOBIAS analysis, more TFs were upregulated than downregulated (Fig. 5f). TF protein levels were unaffected (Fig. 3f and Supplementary Data 2), suggesting that changes in TF binding are not mediated by changes in their concentrations. In contrast to SRCAP, H2A.Z was only slightly decreased in the whole chromatome (Fig. 5f and Supplementary Data 2), suggesting low or SRCAP-independent H2A.Z turnover in regions that are not recovered in ChIP-seq experiments, such as sonication-resistant heterochromatin[60] and repeated regions that are filtered out during processing of ChIP-Seq reads, such as centromeres and telomeres. Because H2A.Z is not only enriched at accessible regions together with SRCAP but also in constitutive heterochromatin where SRCAP is absent, it is expected to see only partial loss of total chromatin-associated H2A.Z upon SRCAP depletion.

We then performed ChIP-seq on selected TFs with altered footprints and/or enrichment in the chromatome. 28.2% of OCT4 and 42% of SOX2 binding sites were more occupied upon IAA treatment, in line with TOBIAS and chromatome analysis (Fig. 5g, h). However, OCT4 was shown to bind H2A.Z nucleosomes[61] and to lose binding upon H2A.Z knockdown[5]. Accordingly, sites bound by OCT4-only lost binding upon H2A.Z swapping with H2A, in contrast to sites bound by both OCT4 and SOX2 (Supplementary Fig. S5f), in line with SOX2-dependent OCT4 binding on canonical nucleosomes[62,63]. Furthermore, SOX2 knockdown (Methods[64]) decreased the number of OCT4 binding sites, and the remaining bound sites were highly enriched for H2A.Z (Supplementary Fig. S5g). Therefore, without SOX2, OCT4 favors binding to H2A.Z over H2A nucleosomes, also explaining the largely intact pioneer activity of OCT4 in the absence of SOX2[34]. NANOG, YY1 and NFY-a also displayed altered genome occupancy in line with TOBIAS and/or chromatome analysis (Fig. 5g), while NANOG protein levels were unchanged after SRCAP depletion (Supplementary Fig. S5h), and OCT4, SOX2 and NANOG sites displayed increased binding in distal regulatory regions (Supplementary Fig. S5i).

Since changes in TF binding were already maximal 2 h after IAA addition (Supplementary Fig. S5c), we reasoned that TFs could drive subsequent changes in chromatin accessibility (Fig. 5a). Loss and gain of TF binding were associated with corresponding changes in chromatin accessibility (Fig. 5i), in line with the high number of pTFs with increased footprints (Supplementary Fig. S5e). Furthermore, loci with increased OCT4 or SOX2 binding displayed more pronounced increased accessibility than loci with stable or decreased binding (Supplementary Fig. S5j). This suggests that alterations in pTF binding drive changes in chromatin accessibility upon SRCAP/H2A.Z depletion.

## DNA methylation changes do not explain changes in TF binding upon SRCAP depletion

Since H2A.Z was proposed to antagonize DNA methylation[26,65], we reasoned that SRCAP loss may increase DNA methylation, which could subsequently mediate changes in TF binding. CpGs were overrepresented in down vs up motifs (Methods and Fig. 5j), and TFs with

decreased footprints included methyl-sensitive NRF1 and ZBTB33[66,67] (Supplementary Data 3).

We quantified changes in DNA methylation at H2A.Z-occupied loci after 4 h of IAA treatment using Enzymatic Methyl Sequencing (EM-seq). Global DNA methylation displayed only minor changes (Supplementary Fig. S5k). While some regulatory elements displayed increased or decreased CpG methylation upon IAA treatment (Supplementary Fig. S5l), their accessibility was unaffected (Supplementary Fig. S5m). Next, we treated SYAT cells with the GSK-3484862 DNMT1 inhibitor for 5 days, which decreases DNA methylation by 80% in PSCs[68] and performed ATAC-seq after 4 h of IAA or control treatment. DNMT1 inhibition only slightly dampened the impact of IAA on increased TF footprints and did not impact decreased TF footprints (Supplementary Fig. S5n), suggesting that changes in DNA methylation play a minor role in regulating TF binding. To determine if TF binding changes could lead to changes in DNA methylation, we identified enriched TF binding motifs in hyper- and hypomethylated H2A.Z peaks at promoters and enhancers. Many TF motifs were enriched in hypomethylated enhancers (Fig. 5k). Our results suggest that acute changes in DNA methylation are mainly secondary to TF activity and play a minor role in further enhancing TF binding at hypomethylated motifs.

## SRCAP and H2A.Z have distinct, complementary roles in transcriptional regulation

To determine H2A.Z-independent functions of SRCAP, we generated a catalytically dead SRCAP protein (cdSRCAP) deficient for H2A.Z deposition by mutating three amino acid residues in its ATPase domain (Supplementary Fig. S6a). We then engineered the SYAT cell line for doxycycline (dox)-inducible expression of either wtSRCAP or cdSRCAP fused to mCherry (Methods), hereafter referred to as wtSRCAP and cdSRCAP cell lines. We induced expression of wtSRCAP-mCherry or cdSRCAP-mCherry for 24 h and treated cells with IAA or control for 4 h. We FACSorted cells with the same average mCherry fluorescence to ensure identical expression levels of wt and cdSRCAP and performed ChIP-seq for H2A.Z (Fig. 6a). cdSRCAP cells displayed lower global H2A.Z enrichment than wtSRCAP cells (Supplementary Fig. S6b), and H2A.Z levels were rescued within <10% of the control condition at 44,769 peaks in wtSRCAP but not cdSRCAP cells (see Methods) (Fig. 6b). In addition, overexpression of cdSRCAP either in CTRL or IAA condition led to a lower enrichment of H2A.Z at H2A.Z peaks, confirming that cdSRCAP is defective in depositing H2A.Z, with a slight dominant negative effect on H2A.Z deposition (Supplementary Fig. S6c).

We then compared transcriptional changes in IAA-treated cells expressing wtSRCAP or cdSRCAP. wtSRCAP compensated transcriptional changes mediated by endogenous SRCAP degradation more than cdSRCAP (Supplementary Fig. S6d). We clustered genes with intronic counts significantly differing between any pair of samples, according to their changes as a function of H2A.Z and SRCAP levels in wtSRCAP or cdSRCAP cells (Fig. 6c). Genes in clusters A and B were negatively regulated by H2A.Z, while genes in clusters D and E were positively regulated by SRCAP. Clusters C and F reflected conflicting effects of H2A.Z and SRCAP on gene regulation, resulting in no significant differences in expression upon SRCAP degradation in SYAT cells (Fig. 6d). Genes upregulated by SRCAP (clusters D and E) were associated with a general cellular function GO term (Supplementary Fig. S6e) and included 70 housekeeping genes, indicating an H2A.Z-independent role of SRCAP in activating ubiquitously expressed genes. H2A.Z played a dominant role in transcriptional repression of lineage-specific genes, confirmed by direct comparison of cdSRCAP versus wtSRCAP-expressing cells in both control and IAA conditions (Fig. 6e, f and Supplementary Fig. S6f, g). We next asked whether the H2A.Z-independent SRCAP function in stimulating transcription is determinant for gene reactivation at mitotic exit. Affected genes from all clusters stimulated by SRCAP were downregulated upon IAA treatment

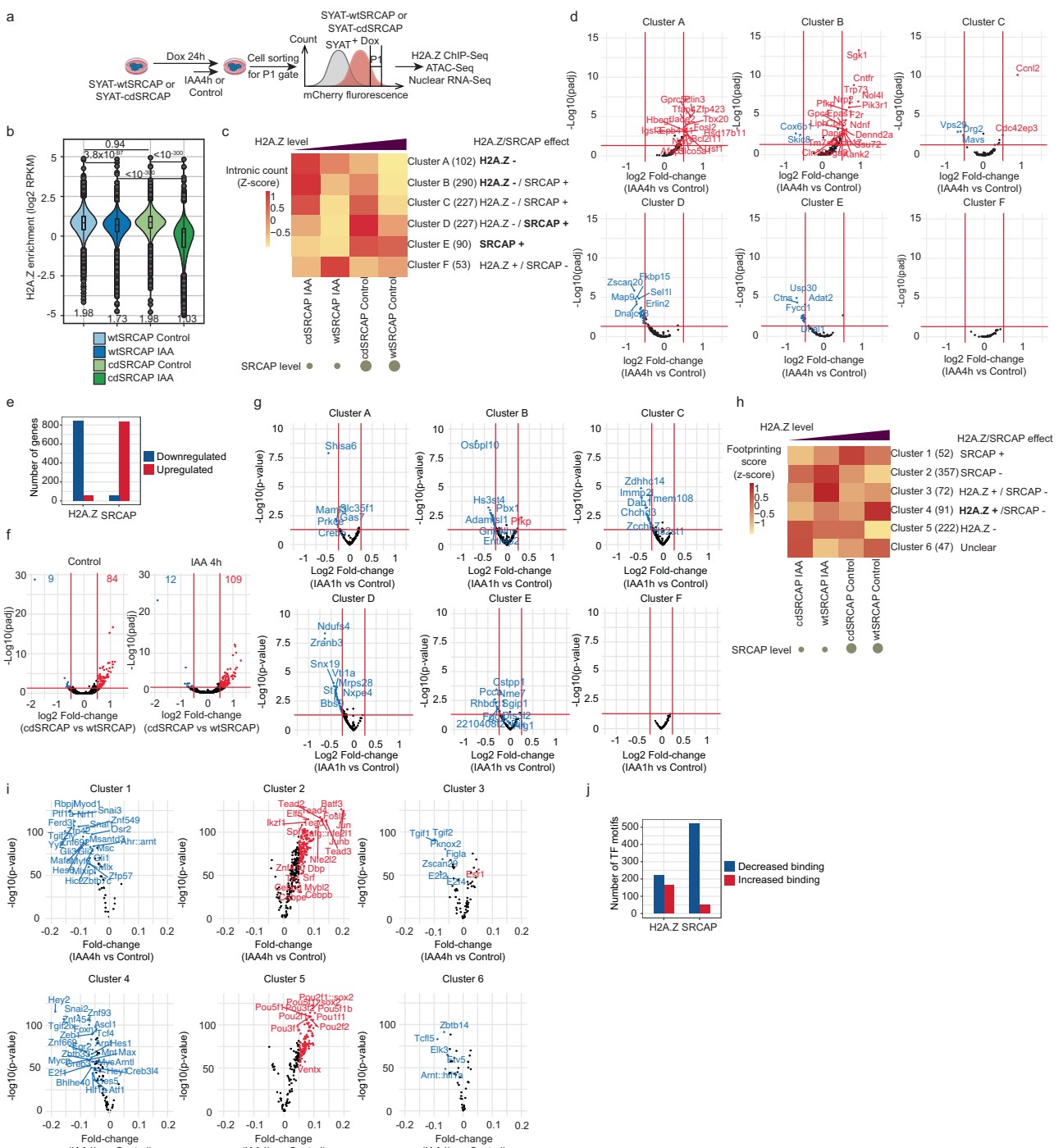

**Fig. 6 | H2A.Z-dependent and independent functions of SRCAP. a** Experimental procedure for dox-induced wtSRCAP and cdSRCAP expression and cell sorting. **b** H2A.Z enrichment in rescued peaks (44769 peaks) upon overexpression of wtSRCAP or cdSRCAP treated with control or IAA for 4 h. Boxes: interquartile range (IQR); whiskers: 1.5xIQR; dots: outlier points. **c** Heatmap of normalized intronic read counts ranked according to H2A.Z and SRCAP levels and k-means-clustered. When SRCAP and H2A.Z display opposite effects, the dominant effect is written in bold. **d** Fold-change of intronic reads from two independent biological replicates and adjusted $p$ value for SYAT cells treated for 4 h with IAA versus control, in the six gene clusters determined in (**c**). Colored dots: genes significantly downregulated (blue) or upregulated (red) with a threshold of |log2FC| >0.5 and adjusted $p$ value <0.05. **e** The number of genes downregulated or upregulated by H2A.Z or SRCAP. **f** Fold-change of intronic reads between cdSRCAP and wtSRCAP and adjusted

$p$ value in control (left) or IAA condition (right). Numbers: genes significantly lower (blue) or higher (red) in cdSRCAP with a threshold of |log2FC| >0.5 and adjusted $p$ value <0.05. **g** Fold-change in intronic reads in the six clusters defined in (**c**) for SYAT cells treated for 1 h with IAA or control and sorted in EG1. **h** Heatmap of TOBIAS footprinting scores ranked according to H2A.Z and SRCAP levels and k-means-clustered. When SRCAP and H2A.Z display opposite effects, the dominant effect is indicated in bold. **i** Fold-change of TF footprinting calculated using TOBIAS in SYAT cells treated for 4 h with IAA versus control for the six clusters determined in (**h**). Colored dots: TF motifs predicted to lose (blue) or gain (red) binding upon IAA treatment with a threshold of |FC| >0.05 < 0.0001. **j** Numbers of TF motifs with binding increased or decreased by H2A.Z or SRCAP. $p$ values for comparison of mean H2A.Z levels in ChIP-Seq were determined by two-sided Welch two-sample $t$-tests. **** $p$ < 0.0001.

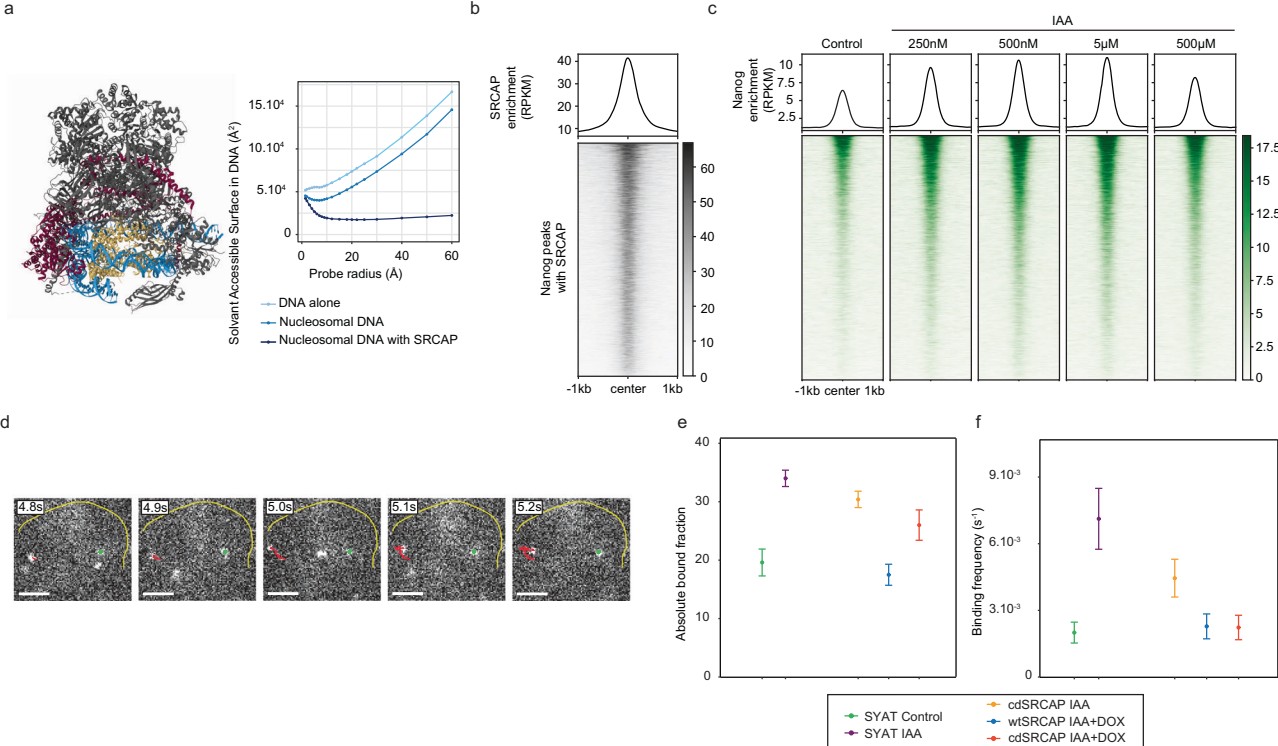

**Fig. 7 | SRCAP decreases NANOG binding by steric hindrance. a** Left panel. Cryo-EM structure of human SRCAP determined by ref. 71, DNA is displayed in blue, histones in yellow, SRCAP subunit in purple, and all other subunits of the SRCAP complex in gray (structure pdb_00008x1c deposited on RCSB PDB, figure created using Mol*[135]. Right panel. Surface of the DNA accessible to molecules of different radii calculated from the Cryo-EM structure. **b** SRCAP enrichment determined by CUT&RUN, centered on NANOG peaks where SRCAP peaks are also detected (8578 peaks). **c** Nanog enrichment at binding sites of both NANOG and SRCAP (8578 peaks) in SYAT cells treated with different doses of IAA for 2 h. **d** Images of single Halo-NANOG molecules in a cell nucleus at the indicated time points, and corresponding tracks (colored). Scale bar: 3 μm. **e** Absolute fraction of Halo-NANOG molecules bound longer than 50 s, calculated from data displayed in Fig. S7f–j, l–n, with the statistics provided there. For CTRL, 2813 tracks in 16 cells, for IAA, 1121 tracks in 26 cells, for cdSRCAP IAA, 4624 cells in 47 cells, for cdSRCAP IAA + DOX, 1623 tracks in 16 cells, for wtSRCAP IAA + DOX, 2571 tracks in 34 cells. Data were represented as calculated mean values ± standard deviation. **f** Binding frequency of Halo-NANOG molecules calculated from data displayed in Fig. S7f–j, l–n, with the statistics provided there. Data, from the same number of tracks as in 7e, are represented as calculated mean values ± standard deviation of 500 resamplings with randomly selected 80% of the data and error propagation.

(Fig. 6g). In contrast, 1 h-IAA treatment in LG1 cells led to transcriptional changes closer to those of longer IAA treatments in asynchronous cells (Supplementary Fig. S6h). This suggests that transcriptional reactivation at mitotic exit is dominated by the stimulatory effect of SRCAP, and that the inhibitory role of H2A.Z comes into play only later during G1 phase.

We then performed ATAC-seq after 24 h of dox and 4 h of IAA treatment in wtSRCAP and cdSRCAP cell lines. Overall changes in chromatin accessibility upon IAA treatment were minor and comparable in both cell lines (Supplementary Fig. S6i). Changes in chromatin accessibility caused by SRCAP depletion were abolished upon wtSRCAP but also cdSRCAP overexpression (Supplementary Fig. S6j), suggesting that these are mostly H2A.Z-independent. We then used TOBIAS to analyze TF footprints in the wtSRCAP or cdSRCAP cell lines in control or IAA conditions. K-mean clustering of TF footprinting scores yielded 6 clusters displaying different dependencies to SRCAP and/or H2A.Z levels (Fig. 6h). We identified two clusters with a predominant role of SRCAP in increasing (cluster 1) or decreasing (cluster 2) TF binding (Fig. 6i), as well as a role for H2A.Z in positive (cluster 3 and 4) or negative regulation (cluster 5) of TF binding at hundreds of motifs (Fig. 6i). Overall, SRCAP strongly prevented TF association with their motifs while H2A.Z had both enhancing and inhibitory effects on TF binding (Fig. 6j). We then compared TF footprinting in rescued peaks in wtSRCAP cells treated with control or IAA, allowing us to determine the impact of different SRCAP levels on TF footprinting independently of H2A.Z levels. Both up and down motifs overlapped substantially with those identified upon IAA treatment of

SYAT cells, confirming the prominent H2A.Z-independent role of SRCAP in regulating TF binding (Supplementary Fig. S6k). SRCAP and, to a lesser extent, H2A.Z tended to hinder TF binding at distal regulatory elements, as seen from the mostly distal localization of the TF motifs in clusters 2 and 5 (Supplementary Fig. S6l). Conversely, both SRCAP and H2A.Z tended to stimulate TF binding at promoters, as seen from the larger fraction of TF motifs located in promoters in their main activatory clusters 1 and 4 (Supplementary Fig. S6l). In addition, SRCAP strongly disfavored pTF association with their motifs (112 SRCAP- vs 5 SRCAP+ motifs) (Supplementary Fig. S6m). Taken together, this suggests that H2A.Z-independent activity of SRCAP plays a major role in restricting pTF binding.

## SRCAP inhibits TF binding by steric hindrance

The catalytically-independent inhibition of TF binding mediated by SRCAP could be caused by physically blocking the access of TFs to their binding sites (steric hindrance) and/or by decreasing TF affinity for their binding sites through chemical modification of chromatin or physical mechanisms acting through deformation of the DNA double-helix[69,70]. Several arguments favor a steric hindrance mechanism. First, the SRCAP complex covers a large fraction on the nucleosome[71–74]. Using a published structure[71], we computed the surface of DNA accessible to molecules, considering either DNA in a free or SRCAP complex-bound nucleosome. For TFs (approximated as pseudo-spheres with radii of 20-30 Å[75,76]), DNA was 2–3-fold less accessible on SRCAP-bound nucleosomes (Fig. 7a). Second, SWR1 complexes remain bound to nucleosomes in vitro for several minutes[50,77]. When

crossing our SRCAP mapping and MNase data, we found that nucleosomes and SRCAP peaks were strongly colocalizing (Supplementary Fig. S7a), confirming that SRCAP mostly occupies nucleosomal DNA in vivo. Third, the motifs of TFs increasing their binding upon SRCAP loss (cluster 2 in Fig. 6) colocalize with SRCAP (Supplementary Fig. S7b). NANOG, which strongly increased its binding upon SRCAP depletion (Fig. 5g, h), displayed maximal SRCAP enrichment at the center of its ChIP-seq peaks (Fig. 7b). Fourth, we reasoned that a steric hindrance mechanism implies its immediate mitigation after SRCAP degradation. After 2 h of IAA treatment, changes in TF footprinting were already maximal (Supplementary Fig. S5c). Furthermore, NANOG ChIP-seq after 2 h of treatment with different doses of IAA (Supplementary Fig. S7c, d) also revealed increased NANOG occupancy in regions initially co-occupied by SRCAP (Fig. 7c).

However, these results do not fully exclude that SRCAP loss may induce rapid changes in local chromatin chemistry or physical properties, which could alter TF binding affinity. The genomic occupancy of TFs depends both on their binding frequency and residence time on specific sites. While alterations of chemical or physical properties of chromatin would alter TF residence time, steric hindrance is expected to decrease TF binding frequency by hiding TF binding sites. To discriminate between these scenarios, we determined the impact of SRCAP on DNA-binding dynamics of NANOG using single-molecule fluorescence tracking (SMT) in live cells (Fig. 7d). We expressed a Halo-NANOG fusion protein at low levels in SYAT cells, cdSRCAP cells – dox, cdSRCAP + dox, and wtSRCAP + dox (Methods). We first determined the overall bound fraction of dye-labeled Halo-NANOG by tracking molecules at 85 Hz[78]. We analyzed the distribution of molecular frame-to-frame jump distances with a multi-component diffusion model[79] (Supplementary Fig. S7e, f), which revealed a higher bound fraction of NANOG in the absence of SRCAP (Supplementary Fig. S7g and Methods). This increase in bound fraction was counteracted by expression of wt or cdSRCAP (Supplementary Fig. S7h–j). Next, we determined the spectrum of residence times of Halo-NANOG by tracking bound molecules at 0.5 Hz using several time-lapse conditions[80] (Supplementary Fig. S7k and Methods). We then focused our analysis on molecules binding longer than 50 s, which are mostly specifically-bound molecules. As expected, the fraction of long-binding molecules was increased in the absence of SRCAP, and this effect was suppressed by expression of wt or cdSRCAP (Fig. 7e and Supplementary Fig. S7m). From the bound fractions and residence times, we calculated the binding frequencies using a kinetic model including diffusing, transiently binding, and stably binding NANOG molecules[81,82] (Supplementary Fig. S7l and Methods). Strikingly, the binding frequency of NANOG increased fourfold upon SRCAP degradation, and this effect was suppressed by expression of wt or cdSRCAP (Fig. 7f and Supplementary Fig. S7n). Interestingly, NANOG residence time was decreased upon SRCAP depletion (Supplementary Fig. S7o), in line with the extension of the NANOG binding landscape to lower affinity regions displaying low accessibility and OCT4/SOX2 occupancy (Supplementary Fig. S7p). These results indicate that increased binding frequency of NANOG drives increased NANOG binding upon SRCAP depletion, in line with SRCAP inhibition of TF binding by steric hindrance.

## Discussion

The transient nature of most DNA-protein interactions is well-established in mammalian systems[83–85]. However, the direct, acute impact of these interactions on chromatin structure and composition is more challenging to characterize. Several studies reported that the recruitment and nucleosome-remodeling activity of the BAF complex occur continuously on a time scale of minutes[34,86,87]. Here we demonstrate that SRCAP depletion for 1–2 h induces marked changes in H2A.Z occupancy at active and bivalent/poised regulatory elements, suggesting that SRCAP also remodels chromatin in a very dynamic manner. The SRCAP-dependent H2A.Z turnover rates at promoters correlating with transcriptional activity is in line with transcription-mediated eviction of full H2A.Z nucleosomes or H2A.Z-H2B dimers[37,88–91], followed by rapid re-incorporation of H2A-H2B and SRCAP-mediated loading of H2A.Z-H2B. The fact that SRCAP binds preferentially to canonical nucleosomes[23] and the correlation of its genomic occupancy with H2A.Z turnover rates (Fig. 1a, h) suggest that the pace of nucleosome eviction sets the rate of SRCAP recruitment to chromatin. Surprisingly, SRCAP-mediated H2A.Z turnover is even faster during mitosis than interphase, despite very low mitotic transcriptional activity[42]. This suggests that SRCAP maintains H2A.Z levels on mitotic chromosomes by competing with transcription-independent H2A.Z eviction mechanisms that remain to be identified.

Despite the initial characterization of SRCAP for its ability to recruit transcriptional coactivators[11–13], its role in gene regulation has become increasingly viewed through the lens of its H2A.Z deposition activity. Our study clarifies the specific and direct functions of SRCAP and H2A.Z in gene regulation. We found no evidence that H2A.Z is directly involved in nucleosome destabilization or inhibition of DNA methylation, but we cannot exclude the existence of indirect mechanisms operating on longer time scales. Our results confirmed that H2A.Z fosters H3K4me3 deposition but has a predominantly repressive impact on transcription, in line with broadening the barrier to elongating RNA Pol II[92], causing enhanced pausing and slowing down pre-initiation complex reloading[20]. Our approach uniquely allowed us to determine H2A.Z-independent functions of SRCAP. We found that SRCAP stimulates the transcription of housekeeping genes, in line with H2A.Z-independent upregulation of housekeeping genes by the *Drosophila* SRCAP-ortholog DOMINO during zygotic genome activation[1]. SRCAP also restricts genome-wide binding of a large number of TFs by blocking access to their binding sites (Fig. 8). Here, we identify steric hindrance as a previously unrecognized mechanism by which chromatin remodelers can restrict transcription factor access to chromatin independently of their enzymatic activity. We cannot rule out that SRCAP may also modulate TF binding through alternative mechanisms, for example, by serving as a docking hub that facilitates the recruitment of TFs or other chromatin modifiers, thereby locally reshaping chromatin architecture. In the future, biochemical approaches probing protein–protein interactions may allow us to shed light on potential direct interaction partners of SRCAP.

We identified sets of genes with an antagonistic transcriptional regulation by H2A.Z and SRCAP. For genes where H2A.Z and SRCAP exhibit compensatory activities, their interplay may stabilize gene expression and contribute to the maintenance of cell identity. Conversely, a prevailing effect of either H2A.Z or SRCAP could permit gene deregulation in response to signaling cues and thereby facilitate lineage commitment.

The dynamic interplay between SRCAP and H2A.Z provides an elegant strategy to maintain the cell identity of PSCs by robust, yet rapidly reversible restriction of both the expression and DNA-binding of lineage-specific TFs. The high degree of evolutionary conservation and the role of SRCAP/H2A.Z in self-renewal of multiple types of stem cells[5–7] suggests their broad involvement in the regulation of stem cell plasticity. We thus propose SRCAP/H2A.Z as a pivotal gene regulatory system operating in parallel to other epigenetic mechanisms to maintain cell identity and plasticity.

## Methods
### Cell culture
CGR8 mouse ES cells (Sigma, Cat#07032901-1VL) and ZHBTc4 mouse ES cells[35] were routinely cultured at 37 °C and 5% $CO_2$ on cell culture-treated dishes coated with 0.1% gelatin (Sigma #G9391-100G) using the following culture medium: GMEM (Sigma #G5154-500ML) supplemented with 10% ES cell-qualified fetal bovine serum (Gibco #16141–079), 1% nonessential amino acids (Gibco #11140–050), 2 mM ʟ-glutamine (Gibco #25030–024), 2 mM sodium pyruvate (Sigma

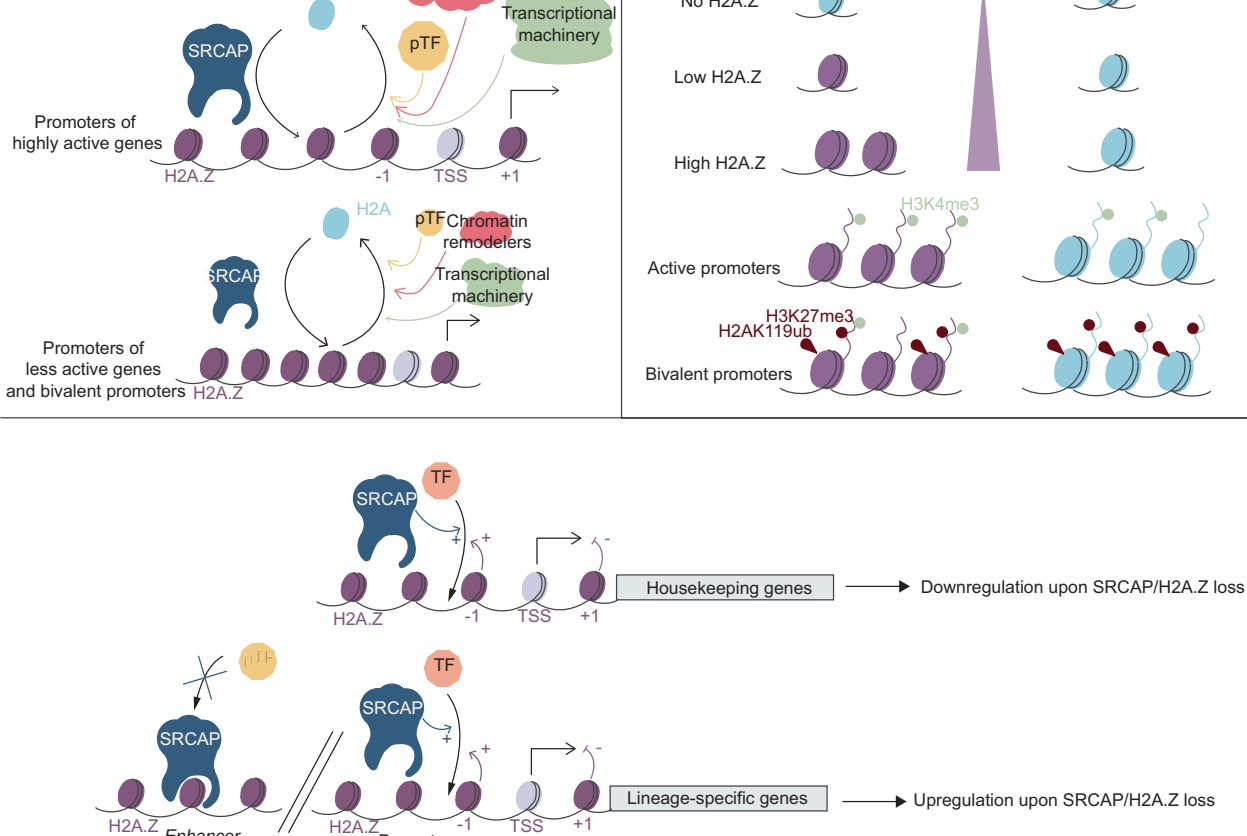

**Fig. 8 | SRCAP regulates H2A.Z turnover, nucleosome properties, TF binding, and transcription in mESCs.**

#S8636-100ML), 100 μM 2-mercaptoethanol (Sigma #63689–25 ML-F), 1% penicillin and streptomycin (BioConcept #4–01 F00-H), in-house produced leukemia inhibitory factor (LIF), CHIR99021 (Merck #361559–5 MG) at 3 μM and PD184352 (Sigma #PZ0181-25MG) at 0.8 μM. Cells were passaged by trypsinization (Sigma #T4049-100ML) every 2 to 3 days. Before all experiments, mESC medium was changed to Serum LIF medium: GMEM (Sigma #G5154-500ML) supplemented with 10% ES cell-qualified fetal bovine serum (Gibco #16141–079), nonessential amino acids (Gibco #11140–050), 2 mM L-glutamine (Gibco #25030–024), sodium pyruvate (Sigma #S8636-100ML), 100 μM 2-mercaptoethanol (Sigma #63689–25 ML-F), penicillin and streptomycin (BioConcept #4–01 F00-H), in-house produced leukemia inhibitory factor (LIF). SYAT cells were cultured routinely with 200 μM Auxinole (MedChemExpress #HY-111444) (Control condition) to avoid basal degradation of SRCAP. For SRCAP degradation, SYAT cells were treated for 1, 2, 4, 6, or 8 h with 500 μM indole-3-acetic acid (IAA, Sigma-Aldrich #I5148). For induction of Oct4 knockdown in ZHBTc4 cells, 1 μg/mL Doxycycline was added for 9 or 12 h prior to cell collection. For DNMT1 inhibition, SYAT cells were treated with 2 μM GSK-3484862 (MedChemExpress #HY-135146) for 5 days before performing ATAC-seq. All used cell lines were tested every 6 months for the absence of Mycoplasma by Eurofins Genomics.

For imaging, cells were plated on Laminin (1/10 dilution of Biolaminin (BioLamina, LN511-0202) in PBS). Prior imaging, medium was replaced by FluoroBrite DMEM (Thermo Fisher, #A18967–01) supplemented with 10% ES cell-qualified fetal bovine serum (Thermo Fisher, #16141079), 2 mM sodium pyruvate (Sigma-Aldrich, #113–24–6), 1% nonessential amino acids (Thermo Fisher, #11140035), 1% penicillin/streptomycin (BioConcept, #4–01F00H), 2 mM L-glutamine (Thermo Fisher, #25030–024), 100 μM 2-mercaptoethanol (Sigma-Aldrich, #63689–25ML-F), and LIF.

## Lentiviral vector production
Lentiviral vectors were produced by transfection of HEK 293T cells with the envelope (psPAX2, Addgene #12260), packaging (pMD2.G, Addgene #12259)[93], and the lentiviral construct of interest using Calcium Phosphate transfection, as described previously[94]. Viral vectors were concentrated 120-fold by ultracentrifugation at $20000 \times g$ for 120 min at 4 °C. About 50,000 cells in 500 μL of medium in a 24-well plate were transduced with 50 μL of concentrated lentiviral vector particles to generate stable cell lines.

## Generation of SYA and SYAT cell lines
The SYA cell line was generated using CRISPR/Cas9-mediated homology-directed repair (HDR). A SRCAP gRNA designed using CRISPOR[95] to target the C-terminal part of mouse SRCAP was added to the pX330-gRNA vector (pX330-gRNA was a gift from Charles P. Lai (Addgene plasmid #158973)) by digestion with BbsI to build pX330_SRCAP. The Ypet_AID_P2A_Hygromycin-resistance cassette was amplified with primers containing 60 bp homology arms corresponding to the carboxy-terminal part of SRCAP, and 2 μg of the amplicons were co-transfected in $1.5 \times 10^6$ CGR8 mouse ES cells together with 2 μg of pX330-SRCAP using Lipofectamine 3000 (Thermo Fisher Scientific #L3000008). Cells were selected by the addition of 150 μg/mL Hygromycin 72 h post-transfection for a week before individual clones were picked. gDNA was extracted using Direct PCR Lysis (Viagen Biotech, #301-C) and incubated 2 h at 55 °C, 30 min at 85 °C, followed by 5 min at 95 °C. Clones were characterized by PCR using

primers annealing to the SRCAP locus and in the knock-in cassette (Table S1), followed by visualization of the Ypet signal intensity and nuclear localization by live imaging using the YFP channel on an IN Cell Analyzer 2200 apparatus (GE Healthcare). One homozygous clone of SYA cells was then transduced with a lentiviral vector containing EF1a-OsTir1-SNAP-P2A-puro and selected with 2 μg/ml of Puromycin (Gibco A11138-03) for a week. To select for cells efficiently degrading SRCAP upon IAA treatment, SYAT cells were treated for 2 h with IAA prior to sorting for Ypet-negative cells. Table 1.

## SRCAP constructs and cell line generation

SRCAP was amplified from cDNA of CGR8 cells in three separate PCR fragments. To generate the catalytically dead mutant, mutations were introduced in the Rev primer of the first fragment and the Fw primer of the second fragment (see Supplementary Table S2 for primer sequences). The three fragments and the pLV-TRE3G-mCherry cut with SalI upstream of the mCherry coding sequence were assembled using In-fusion cloning (Takara Bio # 638947), and the obtained products were transformed into home-made HB101 competent bacteria. Clones were then screened by PCR for the presence of full-length SRCAP and sequenced to verify the presence of the mutated sequence. Correct clones were used to produce lentiviral particles to transduce SYAT cells together with a lentiviral vector expressing rtTA3G and a blasticidin resistance. After selection with 5 μg/mL blasticidin for a week, cells were treated with 1 μg/mL of dox for 24 h to induce expression of SRCAP-mCherry, prior to cell sorting for mCherry-positive cells and isolation of single cells. One clone for wtSRCAP and one clone for cdSRCAP were selected by FACS for similar mCherry levels. Prior to all experiments, wtSRCAP and cdSRCAP were induced with 1 μg/mL doxycycline for 24 h, treated with 500 μM IAA for 4 h or left in 200 μM auxinole (Control) and sorted for similar and high mCherry expression just before performing ChIP-Seq, ATAC-Seq or nuclear RNA-Seq.

## FACS for different cell cycle phases

For mitotic ChIP, $12 \times 10^6$ cells were collected by mitotic shake-off, fixed 10 min with 1% formaldehyde (Thermo Fisher #28908) at room temperature (RT) prior to quenching with 250 mM Tris-HCl pH8. Next, staining with 50 mM Hoechst 33342 (Thermo Fisher Scientific #H3570) was performed for 15 min at RT. Cells were then permeabilized by the addition of ice-cold 70% ethanol and stained with an anti-H3S10P antibody (Merck-Millipore, 05-1336) diluted 1:1000 for 1 h at 4 °C. Samples were stained with Alexa-Fluor 647 anti-rabbit (H + L) secondary antibody (Thermo Fisher Scientific #A-31571) for 45 min at 4 °C. Mitotic cells were selected by FACS as Hoechst-high and H3S10P-positive. For other cell cycle phases, we used DNA content and mitotic degron-based FACS. Mitotic degrons (MDs) have been extensively used in previous publications[34,38,96] as a fusion to fluorescent proteins, inducing their degradation at the metaphase-anaphase transition. As a result, the level of the fluorescent protein remains very low until the middle of the G1 phase, thereby allowing distinction between early G1 and late G1 cells. Here, we used CGR8 cells stably expressing Ypet-MD or SYAT cells stably expressing mCherry-MD, which were stained with Hoechst, followed by sorting according to Hoechst and Ypet/mCherry intensities: Hoechst-low/Ypet-low/mCherry-low cells being in early G1 phase, Hoechst-low/Ypet-high/mCherry-high in late G1 phase, intermediate-Hoechst in S phase and Hoechst-high in G2M phase. For sorting cells at the end of mitosis, SYAT cells expressing mCherry-MD and stained with Hoechst were selected for Hoechst-high/mCherry-low intensities. All FACS were performed on a FACSAria II or a FACSAria Fusion (BD Biosciences).

## Immunofluorescence

SYAT or wt CGR8 cells were plated on 1:10 diluted Biolaminin (BioLamina, LN511-0202) in DPBS (Gibco, 14040091) in 96-well plates with black walls or 3.5 mm dishes. Cells were fixed with 2% Formaldehyde

## Table 1 | List of resources

| Reagent or Resource | Source | Identifier |
|---|---|---|
| Antibodies | | |
| Rabbit anti-H2A.Z | Abcam #ab4174 | RRID:AB_304345 |
| Acetyl-histone H2A.Z (K4,K7,K11) | Thermo Fisher Scientific #PA540095 | RRID:AB_2609565 |
| Rabbit anti-H2A | Abcam #ab18255 | RRID:AB_470265 |
| Rabbit anti-GFP | Abcam #ab290 | RRID:AB_2313768 |
| Rabbit anti-histone H3K4me3 | Abcam #ab8580 | RRID:AB_306649 |
| Rabbit anti-histone H3K27me3 | Abcam #ab195477 | RRID:AB_2819023 |
| Rabbit anti-histone H2AK119ub1 | Cell Signaling Technology #8240 | RRID:AB_10891618 |
| Rabbit anti-YY1 | Cell Signaling Technology #46395 | RRID:AB_2799302 |
| Mouse anti NFYa | Santa-Cruz Biotechnology #sc-17753 | RRID:AB_628018 |
| Rabbit anti-Oct4 | Cell Signaling Technology #5677 | N/A |
| Rabbit anti-Sox2 | Cell Signaling Technology #23064 | RRID:AB_2714146 |
| Rabbit anti-Nanog | Cell Signaling Technology #8822 | RRID:AB_11217637 |
| Rabbit anti-SRCAP | Kerafast #ESL103 | RRID:AB3086743 |
| Biological samples | | |
| Chemicals, peptides, and recombinant proteins | | |
| Indole-3-Acetic Acid (IAA) | Sigma-Aldrich 3I5148 | CAS: 6505-45-9 |
| Auxinole | MedChemExpress 5HY-111444 | CAS:86445-22-9 |
| GSK-3484862 | MedChemExpress 6HY-135146 | CAS: 2170136-65-7 |
| Deposited data | | |
| GSE64825 | De Dieuleveult et al. 2016[123] | N/A |
| GSE48519 | Hon et al. 2014[124] | N/A |
| GSE143737 | Mylonas et al. 2021[20] | N/A |
| Experimental models: Cell lines | | |
| CGR8 (ESC Mus musculus) | Sigma-Aldrich 007032901 | N/A |
| SYA (ESC Mus musculus) | This study | N/A |
| SYAT (ESC Mus musculus) | This study | N/A |
| SYAT-wtSRCAP (ESC Mus musculus) | This study | N/A |
| SYAT-cdSRCAP (ESC Mus musculus) | This study | N/A |
| ZHBTc4 (ESC Mus musculus) | Niwa et al., 2000[35] | RRID: CVCLC715 |
| HEK 293T | ATCC | RRID:CVCL_0063 |
| Oligonucleotides: List provided in Table S1 | | |
| Recombinant DNA | | |
| psPAX2 | Addgene | RRID: Addgene_12260 |
| pMD2.G | Addgene | RRID: Addgene_12259 |
| pX330 | Addgene | RRID:Addgene_42230 |
| pMK-Ypet-AID-P2A-Hygro | This study | N/A |
| pLV-EF1a-OsTir1-SNAP-P2A-Puro | This study | N/A |
| pLV-PGK-Ypet-MD | Deluz et al. 2016[125] | N/A |
| pLV-PGK-mCherry-MD | This study | N/A |
| pLV-rtTA3G-Bsd | Deluz et al. 2016[125] | N/A |

## Table 1 (continued) | List of resources

| Reagent or Resource | Source | Identifier |
|---|---|---|
| pLV-TRE3G-wtSRCAP-mCherry | This study | N/A |
| pLV-TRE3G-cdSRCAP-mCherry | This study | N/A |
| pLV-TRE3G-Halo-Nanog | This study | N/A |
| pLV-pGK-Halo-Nanog | This study | N/A |
| Software and algorithms | | |
| FiJi Version 2.14.0/1.54 f | Schindelin et al., 2012[97] | RRID:SCR_002285 |
| STAR Version 2.7.6a | Dobin et al., 2013[108] | RRID:SCR_015899 |
| Picard Version 2.23.7 | Broad Institute | RRID:SCR_006525 |
| MACS Version 3.0.0a7 | Zhang et al., 2008[110] | RRID:SCR_013291 |
| DeepTools Version 3.5.1 | Ramírez et al., 2016[111] | RRID:SCR_016366 |
| Bedops Version 2.4.41 | Neph et al., 2012[126] | RRID:SCR_012865 |
| BEDtools Version 2.30.0 | Quinlan and Hall, 2010[127] | RRID:SCR_006646 |
| SAMtools Version 1.14 | Li et al., 2009[109] | RRID:SCR_002105 |
| DANPOS2 Version 3.1.1 | Chen et al. 2013[114] | RRID:SCR_015527 |
| Bwa-Meth | Pedersen et al. 2014[128] | RRID:SCR_010910 |
| MethylDackel Version 0.5.3 | https://github.com/dpryan79/MethylDackel | |
| MethylKit Version 1.28.0 | Akalin et al. 2012[119] | RRID:SCR_005177 |
| DeSeq2 Version 1.42.1 | Love M.I et al. 2014[117] | RRID:SCR_015687 |
| edgeR Version 4.0.11 | Robinson et al., 2010[129] | RRID:SCR_012802 |
| limma Version 3.58.1 | Ritchie et al., 2015[130] | RRID:SCR_010943 |
| R Studio Version 2023.12.0 + 369 | | RRID:SCR_000432 |
| Cluster Profiler Version 4.10.1 | Yu et al. 2012[131] | RRID:SCR_016884 |
| ggplot2 Version 3.5.0 | Wickham, 2009[132] | RRID:SCR_014601 |
| ggrepel Version 0.9.5 | Slowikowski K, 2024 | |
| biomaRt Version 2.58.2 | Durinck et al., 2005[133] | RRID:SCR_002987 |
| TOBIAS Version 0.13.3 | Bentsen et al.,2020[56] | |
| SEA Version 5.5.5 | Bailey and Grant, 2021[121] | RRID:SCR_001783 |
| FreeSASA Version 2.1.2 | Mitternacht, 2016[134] | |

the area of the region of interest (ROI) and the intensities for the different channels. For each channel, integrated intensities were calculated by multiplying the mean fluorescence intensity by the area of the ROI. Around 50 cells were quantified for each treatment condition and phase.

### Pluripotency assay and RT-qPCR

About 600 SYAT or CGR8 cells were plated in six-well plates and cultured in GMEM-Serum-2i-LIF for 5 days. Cells were stained with Phosphatase Alkaline staining kit (Merck-Millipore, SCR004) according to the manufacturer's instructions. Colony-forming assays were performed in three biological replicates.

For RT-qPCR experiments, RNAs from three biological replicates were extracted from $3 \times 10^6$ SYAT or CGR8 cells with the Rneasy Plus Mini Kit (Qiagen, #74134). Reverse transcription was performed on 1 μg RNA using oligodT (Thermo Fisher Scientific #SO131) and Maxima H minus First Strand cDNA synthesis (Thermo Fisher Scientific #K1652). The RT reaction was diluted 1/10 prior to qPCR. Primers used to amplify Rps5, Zfp42, Pou5f1, Sox2, Nanog, Esrrb, and c-Myc are detailed in Supplementary Table S1. Plates containing three technical replicates were assembled using an automated Hamilton liquid handling platform and analyzed on a QuantStudio 7 (Applied Biosystem). Relative quantifications were obtained by normalization on Rps5 mRNA levels.

### Western blot

Wt CGR8 or SYAT mESCs in control conditions or treated with IAA for 2, 4, 6, and 8 h were collected and lysed in RIPA buffer (50 mM Tris, pH 8, 1% NP-40, 0.5% NaDoc, 0.1% SDS, 150 mM NaCl supplemented with Protease Inhibitor Cocktail, and 25U of Benzonase (Sigma 70746). About 20 or 40 μg of whole protein extracts were loaded on a 4–20% SDS-Page gel (Bio-Rad) and transferred to an ethanol-activated PVDF membrane. Membranes were blocked in TBS-BSA 5% and incubated with the following antibodies diluted in TBS-Tween 0.1%: SRCAP (Kerafast ESL103), GFP (Abcam ab290), NANOG (Cell Signalling Technology 8822) and GAPDH (Santa-Cruz Biotechnology sc-32233). Membranes were then stained with either anti-mouse or anti-rabbit IgG-HRP and revealed using Clarity Western ECL substrate (Bio-Rad) on an Amersham Imager 680 system (GE Healthcare).

### ChIP-seq

ChIP-seq experiments against H2A.Z, H2A, H3K4me3, OCT4, and SOX2 were performed in duplicates, except for the ChIP-seq against H2A.Z after 1 h of IAA treatment in mitotic versus asynchronous cells, that were performed as single replicates. ChIP-Seq against H3K27me3, H2AK119ub1, NANOG, YY1, and NFYa were performed as single replicates.

For histones and histone post-translational modifications, cells were fixed for 10 min with 1% formaldehyde at RT. For transcription factors, cells were fixed with 2 mM disuccinimidyl glutarate (Thermo Fisher Scientific #20593) for 50 min and in 1% formaldehyde for 10 min. In both cases, fixation was quenched with 250 mM Tris-HCl, pH 8. All fixed cell pellets except mitotic samples were kept on ice and resuspended in Lysis Buffer 1 (50 mM HEPES-KOH, pH 7.4, 140 mM NaCl, 1 mM EDTA, 0.5 mM EGTA, 10% glycerol, 0.5% NP-40, 0.25% Triton X-100 supplemented with Protease inhibitor cocktail (Sigma #P8340-1ML) at 1:100 dilution), incubated 10 min at 4 °C, spun down at 1700 × g, and resuspended in LB1 a second time, spun down and resuspended in Lysis Buffer 2 (10 mM Tris-HCl, pH 8.0, 200 mM NaCl, 1 mM EDTA, 0.5 mM EGTA supplemented with Protease inhibitor cocktail (Sigma #P8340-1ML) at 1:100 dilution), incubated for 10 min at 4 °C, spun down and washed without disturbing the pellet twice with SDS shearing buffer (10 mM Tris-HCl, pH 8.0, 1 mM EDTA, 0.15% SDS supplemented with Protease inhibitor cocktail (Sigma #P8340-1ML) at 1:100 dilution)) and finally resuspended in SDS shearing buffer. For mitotic samples, cell pellets were washed twice in SDS shearing buffer

diluted in DPBS for 30 min at RT. Permeabilization was then performed by addition of a solution of 0.5% Triton X-100 (AppliChem, A1388,0500) for 30 min before addition of blocking solution (BSA 1% (Sigma-Aldrich, A7906) in DPBS) for 30 min. An anti-H2A.Z antibody (Abcam ab4174) was diluted 1:1000 in blocking solution prior to incubation overnight at 4 °C, followed by three washes with 0.1% Tween-20 diluted in DPBS. The secondary antibody Alexa-Fluor-647 Chicken anti-rabbit IgG (H + L) (ThermoFisher Scientific #A-21443), was diluted 1:1000 in blocking buffer and incubated for 1 h at RT. Cells were then washed three times with 0.1% Tween-20 diluted in DPBS and twice with DPBS before addition of DAPI-Fluoromount G (SouthernBiotech, #0100-20). Imaging was performed on an IN Cell Analyzer 2200 apparatus (GE Healthcare) using a 20X magnification objective and the YFP fluorescence channel for Ypet detection, the Cy5 fluorescence channel for AF-647 detection and the DAPI fluorescence channel for DAPI detection. For high magnification imaging of mitotic chromosomes, imaging was performed on a spinning disk microscope (Visitron CSU-W1) with a 60X objective. For image quantification, the Fiji software[97] was used to subtract background and manually determine

and then resuspended in SDS Shearing buffer. Chromatin was sonicated for 20 min at 5% duty cycle, 140 W, 200 cycles or for sorted samples for 10 min at 10% duty, 75 W, 200 cycles on a Covaris E220 focused ultrasonicator. ChIPs were then performed using the ChIP-IT High Sensitivity kit (Active motif, #53040), following manufacturer instructions. About 10 ng of *Drosophila* spike-in chromatin (Active motif, 53083) and 0.5 µg of spike-in antibody (anti-H2Av, Active motif, 61686) were processed together with mouse ES cell chromatin (1.5 µg for ChIP-seq on mitotic cells and matching asynchronous cells, 650 ng for ChIP-Seq after 1 h of IAA treatment, 4 µg for the other H2A.Z and H2A ChIP-seq, 2 µg for H3K4me3, H3K27me3, and H2AK119ub1 and 5 µg for ChIP-seq against TFs) as an internal experiment calibrator[98]. Libraries were prepared with the NEBNext Ultra II DNA Library Prep Kit (NEB #E7645) and sequenced on an Illumina NextSeq 500 using 75-nucleotide read-length paired-end sequencing.

## CUT&Tag
CUT&Tag experiments were performed in duplicates using the CUT&Tag-IT Assay Kit (Active Motif #53160) and following manufacturer instructions. Briefly, 200,000 SYAT cells were bound on Concanavalin A beads prior to incubation overnight with 1:50 anti-GFP rabbit antibody (Abcam #ab290). Next, cells were incubated with a secondary antibody, pA-Tn5-transposomes, and washed according to the manufacturer's instructions. Transposition was subsequently performed for 1 h at 37 °C, and transposed DNA was purified prior to Library Preparation and size exclusion with SPRI beads using a 0.5X ratio while keeping the unbound fraction and 1X ratio on the left side, taking the bound fraction. Libraries were sequenced on an Illumina NextSeq 500 using 75-nucleotide read-length paired-end sequencing.

## CUT&RUN
One replicate of CUT&RUN was performed on 100,000 SYAT cells using an anti-SRCAP antibody (Kerafast #ESL103) following the protocol used in ref. 32.

## ATAC-seq
All ATAC-seq experiments were performed in biological duplicates. About 50,000 cells were collected either directly after trypsinization or after sorting as described above, and subjected to ATAC-seq as described previously[99]. All centrifugation steps were done at 800×g at 4 °C. Briefly, cells were centrifuged for 5 min and washed with cold PBS, then centrifuged for 5 min and resuspended in cold lysis buffer (10 mM Tris-HCl, pH 7.4, 10 mM NaCl, 3 mM MgCl$_2$, and 0.1% NP-40), and centrifuged for 10 min. Subsequently, nuclei were resuspended in a solution of 0.5 mM Tn5 (in-house preparation according to ref. 100) in TAPS-DMF buffer (10 mM TAPS-NaOH, 5 mM Mgcl$_2$, and 10% DMF) and incubated for 30 min at 37 °C. DNA was immediately purified using column purification (Zymo #D4004) and eluted in 10 µl nuclease-free water. Transposed DNA was amplified in a solution containing 1X NEBNext High Fidelity PCR Master Mix (NEB #M0541L), 0.5 µM of Ad1.1 universal primer, 0.5 µM of Ad2.x indexing primer, and 0.6x SYBR Green I (Thermo Fisher Scientific #S7585) using 72 °C for 5 min, 98 °C for 30 s, and five cycles of 98 °C for 1 s, 63 °C for 30 s, and 72 °C for 60 s. About 10 µl of amplified DNA was analyzed by qPCR to determine the total number of cycles to avoid amplification saturation and accordingly amplified with additional three to seven cycles at 98 °C for 10 s, 63 °C for 30 s, and 72 °C for 60 s. DNA was purified using column purification (Zymo #D4004) and size-selected by taking the unbound fraction of 0.55X AMPure XP beads (Beckman Coulter #A63880) followed by the bound fraction of 1.2X AMPure XP beads. Libraries were sequenced on an Illumina NextSeq 500 using 75-nucleotide read-length paired-end sequencing.

## Nuclear RNA-seq
All RNA-Seq experiments were performed in biological duplicates. Following cell collection and wash in PBS, cell pellets were resuspended in 500 µL Nuclear RNA lysis buffer (10 mM Tris, pH 7.4, 10 mM NaCl, 3 mM MgCl$_2$, 1% BSA (Merck, #B9000S), 0.1% Tween (Fisher Scientific, #10113103), 1 mM DTT (Merck, #646563), 0.5 U/µl Protector RNase Inhibitor (Merck, #3335399001), 0.1% NP-40, 0.01% Digitonin (Invitrogen, #BN2006)) and incubated 5 min on ice. Then 500 µL Nuclear RNA wash buffer (10 mM Tris, pH 7.4, 10 mM NaCl, 3 mM MgCl$_2$, 1% BSA (Merck, #B9000S), 0.1% Tween (Fisher Scientific, #10113103), 1 mM DTT (Merck, #646563), and 0.5 U/µl Protector RNase Inhibitor (Merck, #3335402001) were added before spinning down cells at 300×g 4 °C for 5 min. Two more washes with Nuclear RNA wash buffer were performed before proceeding to RNA extraction using the Rneasy Plus Micro Kit (Qiagen, #74034) with the addition of an on-column DNAse digestion (Qiagen, #79254). Sequencing libraries were prepared by the EPFL Gene Expression Core Facility using the NEBNext Ultra II Directional RNA Library Prep Kit for Illumina (NEB, #E7760S), and sequenced on a NovaSeq6000 using 75-nucleotide read-length paired-end sequencing.

## MNAse-seq
All MNAse-Seq experiments were performed in biological triplicates according to the following protocol[101]. Briefly, Micrococcal Nuclease S7 (Roche #10107921001) was resuspended to 15 U/µL in 20 mM Tris-HCl, pH 7.5, 50 mM NaCl, and 50% Glycerol. For high digestion conditions, MNase was diluted at 100 U/mL in MNase Digestion buffer (50 mM Tris-HCl, pH 8, 150 mM sucrose, 50 mM NaCl, and 5 mM CaCl$_2$). For low digestion conditions, MNase was diluted to 6.25 U/mL in MNase Digestion Buffer. SYAT cells were treated for 4 h with 500 µM IAA or 200 µM Auxinole before collection. About $5 \times 10^6$ cells were used for each MNase digestion condition. After cell collection, cell pellets were resuspended in 50 µL Resuspension Buffer (35 mM HEPES, pH 7.4, 150 mM sucrose, 80 mM KCl, 5 mM KH$_2$PO$_4$, 5 mM MgCl$_2$, 0.2% NP-40). Then, 500 µL of MNase solution were added to the cells and incubated for exactly 10 min at 37 °C before inactivation of MNase by addition of 11 µL of 0.5 M EDTA to the reaction. Next, 550 µL of Lysis Buffer (50 mM Tris-HCl, pH 8, 1% SDS, and 10 mM EDTA) was added, and samples were incubated on ice for 10 min. After the addition of 1.1 mL DNAse-free water, samples were digested for 2 h at 37 °C with RNase (50 µg per reaction). Finally, 3200 U of Proteinase K were added, and digestion was performed at 56 °C for 2 h. Digested DNA was purified using a DNA Purification kit (Active Motif #58002). Then, 500 ng of DNA for the High digestion condition and 1500 ng of DNA for the Low digestion condition were loaded on a 1.5% agarose gel, and the band below 200 bp was cut out and gel-purified. Library preparation was carried out on the purified DNA using NEBNext Ultra II DNA Library Prep Kit (NEB #E7645) and sequenced at 100 million reads per sample on a NovaSeq6000 using 75-nucleotide read-length paired-end sequencing.

## Enzymatic methyl-sequencing
EM-Seq experiments were performed in biological duplicates. Genomic DNA was extracted using a DNeasy Blood and Tissue kit (Qiagen #69504) and repurified using a DNA Clean and Concentrator-5 Kit (Zymo #D4004). Next, 200 ng of purified genomic DNA were sonicated for 90 s, 10% duty, 140 W, 200 cycles on a Covaris E220 focused ultrasonicator and were subsequently processed using NEBNext Enzymatic Methyl-Seq kit (NEB #7120) following the manufacturer's protocol. Libraries were sequenced at 200 million reads per sample on a NovaSeq6000 using 75-nucleotide read-length paired-end sequencing.

## Whole proteome and chromatome analysis by mass spectrometry

The whole proteome and chromatome sample preparation, liquid chromatography followed by tandem mass spectrometry (LC-MS/MS) and data analysis were performed as previously described in ref. 59, with a few adjustments: for the whole proteome-chromatome-matched sample collection, $7 \times 10^6$ embryonic stem cells were harvested per condition and replicate. The cells were washed with PBS and split into two parts: $2 \times 10^6$ cells for whole proteome analysis and $5 \times 10^6$ cells for chromatome analysis. Liquid chromatography was performed using the EvosepONE system with a 15 cm × 150 µm column containing 1.9 µm C18 beads (PepSep). Each sample (200 ng) was loaded onto Evotips according to the manufacturer's protocol. The gradient length was 44 min (30SPD). The column temperature was maintained at 40 °C, and electrospray ionization was achieved using a stainless-steel emitter (30-µm inner diameter) at 2.2 kV. Mass spectrometry was conducted on an Orbitrap Exploris™ 480 mass spectrometer (Thermo Fisher Scientific) in data-independent acquisition (DIA) mode. MS1 acquisitions were performed as previously described, using an Orbitrap resolution of 120,000 with a scan range of 350–1400 m/z and a maximum injection time of 45 ms. For the MS2 acquisition, adjustments were made to accommodate the shorter LC gradient. Specifically, 49 scan events were performed with isolation windows of 13.7 m/z at an Orbitrap resolution of 15,000. The normalized AGC target was set to 3000%, and the maximum injection time was 22 ms. The precursor mass range was maintained at 361–1033 m/z.

The list of TF was determined using a list of proteins associated with the GO terms DNA-binding transcription factor activity (GO:0003700) and Sequence-specific DNA binding (GO:0043565).

## Single-molecule imaging

We performed single-molecule imaging on SYAT, wtSRCAP and cdSRCAP cells overexpressing Halo-Nanog under the control of doxycycline-inducible TRE3G or constitutive pGK promoters, respectively. We seeded cells on 35-mm glass-bottom dishes (Ibidi, 81158) previously coated with a 1:10 dilution of Biolaminin in PBS. Seventy-two hours prior to imaging, we added 200 µM Auxinole to the culture medium for all cell lines. For SYAT cells, we induced expression of Halo-Nanog by adding 0.5 µg/ml Doxycycline, 24 h before imaging. Staining was performed right before Doxycycline induction, 24–25 h prior to imaging. We stained SYAT cells for single-molecule tracking with 100 or 25 pM HTL-SiR (kindly gifted by Kai Johnsson, MPI Heidelberg), respectively, for 85 Hz or 0.5 Hz movies. To visualize the nucleus, we performed additional staining with 62 nM HTL-TMR (Promega, G8252). After staining, cells were incubated for 30 min, then washed twice with PBS and incubated overnight in medium containing 0.5 µg/ml Doxycycline. To induce SRCAP degradation, we washed cells twice with PBS to remove residual auxinole and treated SYAT cells with 500 µM IAA 4 h before imaging. Cells were then washed three times with PBS and imaged in OptiMEM. We induced the expression of wtSRCAP and cdSRCAP cells by adding 1 µg/ml Doxycycline 24 h before imaging. cdSRCAP cells were stained with 1 pM or 5 pM HTL-SiR, respectively, for 85 Hz or 0.5 Hz movies. wtSRCAP cells were stained with 50 pM HTL-SiR for both types of movies. The incubation time was in all cases 15 min. We subsequently washed cells three times with PBS and imaged in OptiMEM.

Single-molecule tracking experiments were conducted on a custom-built inverted microscope with highly inclined laminated optical sheet (HILO) illumination[102]. The fluorophores were excited with a 638 nm laser (Omicron Luxx Laser, Omicron-Laserage Laserprodukte Gmbh), which was spatially cleaned by a single-mode fiber (kineFLEX HPV, Excelitas Technologies Corp.). The beam was further expanded to a FWHM diameter of around 6 mm with a fiber-coupler (60FC-T-4-M50L-0, Schäfter + Kirchhoff GmbH) and its profile was altered from a Gaussian to a flattop via a diffractive beam shaper

(piShaper 6_6_VIS, AdlOprica GmbH). The laser was focused into the back-focal plane of the objective (CFI SR HP Apochromat TIRF 100xC Öl, Nikon GmbH) over an achromatic lens (AC254-200-A, Thorlabs GmbH) and a dichroic mirror (F73-537, AHF Analysetechnik). The fluorescence light was filtered by a combination of an emission and a notch filter (F67-532 and F40-074, AHF Analysetechnik) and detected on a sCMOS camera (Prime BSI, Teledyne Scientific Imaging GmbH). Pixels on the sCMOS camera were binned 2 × 2, resulting in an effective pixel size of 130 nm. The sample was mounted in a stage incubator (Inkubator XS 2000, PeCon GmbH) at 37 °C and 5% $CO_2$ and was controlled in x-y-z direction with two stacked piezo stages (M-686 and M-562, Physik Instrumente GmbH).

To determine the diffusion coefficients and bound fraction of Halo-NANOG, we performed tracking of molecules with a camera exposure time of 10 ms. Each cell was imaged only once. SYAT cells were imaged with a 638 nm laser at 650 µW for 2500 consecutive frames. cdSRCAP and wtSRCAP cells were selected for low or zero Ypet signal, and additionally for high mCherry signal in the case of + dox measurements, then imaged with a 638 nm laser at 780 µW for 2000 consecutive frames.

To determine the chromatin residence time of Halo-NANOG, we performed measurements with a camera exposure time of 500 ms at continuous illumination (continuous illumination) or alternating between a single illuminated frame and a fixed dark time of 10 s (time-lapse illumination). For both illumination schemes and for all the cell lines, we imaged with a 638 nm laser at 30 µW. Each cell was only imaged once. Global analysis of both illumination schemes allowed us to differentiate between dissociation of molecules and photobleaching or tracking errors, respectively[80].

## Analysis of single-molecule movies

Analysis of single-molecule tracking data was performed using TrackIt[79]. For 85 Hz movies, spots were detected with a threshold factor of 1 and a tracking radius of 4 px. For 0.5 Hz movies, we selected threshold factors of 1.0, 1.4, and 1.3, respectively, for the SYAT, the wtSRCAP and the cdSRCAP cell line. The tracking radii were 3 px for the continuous and 5 px for the time-lapse illumination, for all cell lines. The minimum tracking length was two frames for all conditions and cell lines, allowing for one gap frame if the track lasted longer than two frames. We determined diffusion coefficients and fractions for Halo-NANOG by fitting the cumulative distribution of squared jump distances obtained from the 85 Hz movies with a three-component diffusion model[78]. To estimate the error for every diffusion coefficient, we repeated the analysis 400 times, with 80% of randomly selected jump distances. The overall bound fraction of Halo-NANOG was defined as the amplitude A1 of the slowest apparent diffusion component, originating from the localization error and slow chromatin movement. The other two components A2 and A3 accounted for anomalous diffusion in the nucleus[103,104]. Next, we determined the dissociation rates of Halo-NANOG from chromatin using GRID[80]. GRID generates a spectrum of dissociation rates by inverse Laplace transformation of the distribution of fluorescence survival times for Halo-NANOG-bound SiR. An estimate of the error associated with each dissociation rate was given by repeating the GRID analysis 500 times with 80% of randomly selected data from the survival time distributions. The residence times of Halo-NANOG were calculated as the inverse of the dissociation rates. For further analysis, we focused on the longest residence times, as this is typically associated with the specific function of a transcription factor[105–107]. The absolute fraction of long-bound molecules was calculated as the product of the overall bound fraction with the amplitude of the state spectrum associated with the longest binding time. The binding frequencies for Halo-NANOG were calculated as the inverse of the search times determined from bound fractions and dissociation rates as described[81,82].

## ChIP-Seq and CUT&Tag analysis

Newly-generated ChIP-sequencing libraries, as well as ChIP-Seq for BRG1, Ep400, CHD4, H3K27ac, H3K4me1, H2A.Z.1, and H2A.Z.2 that were reanalyzed, were mapped to the mm10 version of *Mus musculus* genome or the version BDGP6 of *Drosophila melanogaster* genome, using STAR[108] with parameters --alignMatesGapMax 2000 --alignIntronMax 1 --alignEndsType EndtoEnd --outFilterMultimapNmax 1. For NANOG ChIP performed on SYAT treated with different doses of IAA for 2 h, raw sequencing files from two replicates were pooled prior to mapping to the genome to increase sequencing depth. Duplicated reads and reads not mapping to chromosome 1–19 and X or Y were removed using Picard (Broad Institute). For the generated Drosophila alignments bam file, the number of reads was determined, and normalization factors were calculated to bring all the *Drosophila* bam files to the same number of reads. These normalization factors were then applied to downsample alignment files mapped on the mouse genome using SAMTools[109]. Peak calling was then performed on these downsampled bam files using MACS2[110] with settings -f BAMPE -g mm, regions contained in mm10 blacklist from ENCODE consortium being excluded. Scores were determined from downsampled bam files using the bamCoverage function from DeepTools[111] with setting --normalizeUsing RPKM.

For score comparison, a bed file regrouping all peaks called in at least one condition was generated, and enrichment was calculated in these defined regions using the multiBigwigSummary function from DeepTools. For classification based on differences of peak amplitude between two time points of IAA treatment or with control, peaks were considered to be different in amplitude if there was a variation of at least 20% of enrichment score in peaks (RPKM-normalized) for H2A.Z, H3K4me3, H3K27me3, and H2AK119ub or at least 30% for YY1, NFYa, OCT4, SOX2, NANOG, and H2A.Z in mitotic and asynchronous cells treated with IAA for 1 h or control.

For H2A.Z ChIP-Seq performed across the cell cycle, promoter and enhancer clusters were determined by k-means clustering of the log2 fold-change of the signal between G2 and M phase together with log2 fold-change of the signal between M and EG1 phases, using the plotHeatmap function of DeepTools.

For H2A.Z ChIP in wtSRCAP or cdSRCAP-expressing SYAT cells, peaks were considered as rescued if the difference between IAA and Control enrichment scores was higher (with a 10% margin) in cdSRCAP compared to wtSRCAP.

In boxplots, the edges of the box represent the first and third quartiles (Q1 and Q3), the bar represents the median value and the dots are the outlier values (lower than Q1-1.5xInterquartile-range, higher than Q3 + 1.5xInterquartile-range).

## Establishment of reference bed files

The list of promoters was determined from a list of TSS in mm10 genomes (obtained from refTSS from the UCSC Genome Browser) that was enlarged from 1 kb upstream of the TSS and 500 bp downstream. To determine the list of active promoters, this list was crossed with H3K4me3 peaks without any repressive histone marks, for bivalent promoters with peaks from H3K4me3 and H3K27me3 ChIP-Seq and for inactive promoters with H3K27me3 peaks that overlapped with no activating histone marks. To identify active enhancers, promoter regions were excluded, and then H3K27ac peaks were crossed with H3K4me1 peaks. Poised enhancers were determined by crossing H3K4me1 enrichment with H3K27me3 enrichment. Regions outside promoters and without H3K4me1 enrichment were classified as inactive enhancers. For analysis using all enhancers grouped together, the list from the FANTOM5 consortium was used, or alternatively, we used the list from Enhancer Atlas[112] established in J1 mESCs when promoter-enhancer loop information was required. In this database, enhancers were annotated using accessibility data, enrichment for H3K4me1, H3K27ac, and p300, and expression of bi-directional eRNA from GRO-

Seq data, in different species and cell types. We used the list of enhancers obtained for J1 mouse ESCs. In the Enhancer Atlas, interactions between a given annotated enhancer and promoters are derived from RNA Pol II-based ChIA-PET data. To determine the number of promoter-enhancer loops, the number of enhancers interacting with the promoter from one given gene was computed using the Enhancer Atlas list, separating upregulated genes from downregulated genes upon SRCAP depletion. Additionally, we also reused the Supplementary Data S9 from[113] presenting all promoter-enhancer loops detected in Micro-C data from wt mESCs together with their strength score, filtering these data for promoters of downregulated and upregulated genes after SRCAP depletion.

The list of TSS was enlarged from 20 bp upstream and 20 bp downstream of the TSS to determine the position of the TSS-nucleosome. Then the 400-pb upstream of this position and 400-pb downstream were considered as regions of positioning for the −1 or +1 nucleosomes, respectively.

Enhancer RNAs were defined as follows: the middle position of enhancers from the J1 enhancer list was determined, and RNA reads falling into 3 kb upstream and downstream were extracted, except those falling into exons and long non-coding RNA (list taken from LncRBase, http://bicresources.jcbose.ac.in/zhumur/lncrbase/).

H2A.Z.1-only, H2A.Z.2-only, and shared peaks were determined by intersecting bed files generated from ChIP-Seq reanalysis of data from ref. 20 using the bedtools intersect function. OCT4-dependent, SOX2-dependent, and co-dependent regions were extracted from a previous study[34].

## MNase-seq analysis

MNAse-seq libraries were mapped and filtered as described for ChIP-Seq. Nucleosome positions were determined using the dpos function from DANPOS2[114]. Enrichment scores were computed using the bamCoverage function from DeepTools[111] with the setting --normalizeUsing RPKM. For all nucleosomes determined using DANPOS, the H2A.Z enrichment score was calculated, and nucleosomes were classified into "No H2A.Z" (3% lowest H2A.Z scores), "low H2A.Z" (first quartile except "No H2A.Z"), "High H2A.Z" (Upper quartile) and "Intermediate H2A.Z" (remaining nucleosomes). These nucleosomes were further classified depending on whether they were in promoters or enhancers, and according to the activity of these regulatory regions. Occupancy data were extracted from a comparison of MNhigh and MNlow datasets using DANPOS. For MNase sensitivity, MNhigh and MNlow scores were determined for all nucleosomes using the multiBigwigSummary function from DeepTools and the MNlow over MNhigh ratio was calculated. For fragment length analysis, subsets of bam files were generated according to H2A.Z enrichment and regulatory regions using the samtools view function with parameters -b -h -L. The lengths of the fragments in the bam files were then determined with the samtools view -f66 | cut -f 9 command line.

## Nuclear RNA-seq analysis

Nuclear RNA-Seq libraries were mapped to the mm10 mouse genome using STAR. Exon coordinates were gathered from GENCODE annotation from vM25 of the UCSC Genome Browser. Introns' coordinates were determined by subtracting exon coordinates from full-length gene bodies using a script generated in a previous study (https://github.com/charitylaw/Intron-reads[115]). Gene counts were then determined using the featureCounts function[116] from the R package Rsubread. Differential enrichment analysis as well as normalization of results using the lfcShrink function with type ="normal" were performed using DeSeq2[117]. Read counts were normalized on intron length and in transcript per million (TPM) for analysis of absolute RNA levels in the control condition.

The bivalent genes list was determined from the list of bivalent promoters. The list of housekeeping genes was taken from a previous

study[118]. Finally, the list of genes associated with pluripotency, mesoderm commitment, ectoderm commitment and endoderm commitment were compiled from Datanode using the terms "Mechanism associated with pluripotency (WP1763)", "Mesodermal commitment pathway (WP2857)", "Ectoderm differentiation (WP2858)" and "Endoderm differentiation (WP2853)", respectively.

For k-means clustering of gene expression between wtSRCAP and cdSRCAP treated with IAA or control, intronic reads count for genes affected in at least one condition were normalized by DeSeq2 using the median of ratios method. Then, for each gene, the z-scores across samples were computed. K-means clustering was next performed using the pheatmap function from pheatmap R package (https://CRAN.R-project.org/package=pheatmap) using a fixed number of six clusters.

### EM-seq analysis
Libraries were mapped using bwa-meth from Bwa (from Brent Pedersen, https://github.com/brentp/bwa-meth). CpG methylation was then calculated using MethylDackel (https://github.com/dpryan79/MethylDackel) before analysis of Differential methylation using MethylKit[119]. CpG methylation was computed for H2A.Z peaks located in promoters and enhancers, and if the percentage of methylation was increased or decreased by 5%, regions were considered as hypermethylated or hypomethylated, respectively.

### ATAC-seq analysis
ATAC-Seq libraries were mapped and filtered as described for ChIP-Seq. The alignment files were subsequently downsampled so that all files contain the same total number of reads. Accessible peaks were then called, and scores were computed as described above for ChIP-Seq. Differentially-accessible regions were determined among peaks that had been called in at least one condition, for scores that varied from at least 30% between control and 4 h of IAA treatment.

### TOBIAS analysis
For TOBIAS analysis, alignment files from ATAC-Seq were normalized with trimmed mean of M values (TMM) using calcNormfactors from edgeR with the TMM method, followed by downsampling using SAMTools. Then, a correction of Tn5 bias was performed using TOBIAS ATACCorrect in all accessible regions in SYAT treated with control or IAA, excluding mm10 ENCODE-blacklisted regions. Footprints were then assessed with TOBIAS FootprintScore in the same regions. Finally, conditions were compared using TOBIAS BINDetect on the same regions or H2A.Z-rescued regions when specified, using the Jaspar2022[120] non-redundant core list of motifs to which the Nanog motif (UN0383.1) was manually added. The significance threshold of p < 0.0001 and |FC| >0.5 for TOBIAS analysis were determined so that changes in wt CGR8 treated with IAA were very mild and not overlapping with the motifs that we found affected in SYAT cells treated with IAA.

The list of pioneer transcription factors (pTFs) (Supplementary Table S2) was compiled from the list published by ref. 57 in their Supplementary Table 1 and ref. 58 in their supplementary Table 2 and Supplementary Data 4. In these studies, authors classified TFs that were able to bind nucleosomal DNA and induce the opening of bound regions at least in one cell type as pTFs.

For the determination of motifs containing a CpG, power weight matrices (PWM) for all motifs were collected, and the CpG score was calculated as follows.

For a motif with N positions:

$$CpG\ score = \sum_{i=1}^{N-1} P(C)_i \times P(G)_{i+1} \times (Weight_i + Weight_{i+1}) \quad (1)$$

With for each position i,

$$P(C)_i = \frac{Score(C)_i}{Score(A)_i + Score(T)_i + Score(G)_i + Score(C)_i} \quad (2)$$

And

$$Weight_i = \frac{Score(A)_i + Score(T)_i + Score(G)_i + Score(C)_i}{mean(Score_{matrix})} \quad (3)$$

$mean(Score_{matrix})$ being the mean of all the scores present in the position weight matrix.

Then, motifs with a CpG score >6 were considered as containing at least one CpG.

The P/E score was calculated for each motif as the likelihood that each motif occurrence is present in a promoter rather than an enhancer. For each occurrence of a motif present in a promoter, the score is 1, if it is in an enhancer, it is 0. These scores are then averaged over all motif occurrences to give the P/E score of each TF motif.

For k-means clustering of TF predicted binding scores in wtSRCAP and cdSRCAP treated with IAA or control, TF binding scores were extracted from TOBIAS results for all motifs of the Jaspar2022 database and the Nanog motif. Z-scores were then calculated across samples for each motif and k-means-clustered using the pheatmap function with a fixed number of six clusters.

### Motif and Gene Ontology (GO) enrichment
Motif enrichment was performed using the SEA algorithm[121] from the MEME suite using the Jaspar non-redundant core database as motif database and the default parameters for the other settings. GO enrichment analysis was performed using GOnet[122] with the default parameters and either biological process or molecular function filters for GO terms.

### Published datasets
BRG1, Ep400, CHD4 ChIP-Seq raw datasets were obtained from GSE64825[123]. H3K27ac, H3K4me1, and H3K27me3 datasets were obtained from GSE48519[124]. H2A.Z.1 and H2A.Z.2 ChIP raw datasets were obtained from GSE143737[20]. SRCAP CUT&RUN data in wt mESCs were obtained from GSE256314[32].

### Reporting summary
Further information on research design is available in the Nature Portfolio Reporting Summary linked to this article.

## Data availability
All sequencing data were deposited at NCBI Gene Expression Omnibus (GEO), accession number GSE269310 [https://www.ncbi.nlm.nih.gov/geo/query/acc.cgi?acc=GSE269305]. Proteomics data were deposited at ProteomeXchange, accession number PXD052934. Source Data have been deposited at Zenodo, under https://doi.org/10.5281/zenodo.18161555.

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

## Acknowledgements

We thank the Gene Expression, Flow Cytometry, BioImaging and Biomolecular Screening core facilities at EPFL. We thank Dr Matthias Mann for kindly providing access to MS instrumentation, and to Dr Igor Paron and Dr Tim Heymann for outstanding MS technical assistance. We thank Dr Martin Pacesa for help in designing cdSRCAP. We thank Dr Ludovica Vanzan, Dr Nadine Vastenhouw and Dr Nicolas Thomä for their input. We thank Dr Marcel Tisch and Dr Pauline Franz for respectively sharing the drawing of stem cells colonies and nucleosomes. We are thankful for funding by the Swiss National Science Foundation

grant#310030_184782 and# 310030_212197 to DMS. HL and EU are funded by the Deutsche Forschungsgemeinschaft (DFG, German Research Foundation, 213249687 - SFB1064 to H.L. (A17)). E.U. gratefully acknowledges the International Max Planck Research School for Molecular Life Sciences (IMPRS-LS) and the Research Training Group 1721 (RTG 1721) for training and support. The work from S.D.L., D.A., and J.C.M.G. was funded by the Deutsche Forschungsgemeinschaft (DFG, German Research Foundation no. 468578170 and CRC 1506 C05 no. 450627322 to J.C.M.G.).

## Author contributions

Conceptualization, A.T. and D.M.S.; Methodology, A.T., E.U., S.D.L., and D.M.S.; Formal analysis, A.T., E.U., S.D.L., and D.M.S.; Investigation, A.T., E.U., S.D.L., and C.D.; Instrumentation (single-molecule imaging microscope), D.A.; Resources, D.M.S., H.L., and J.C.M.G.; Writing original manuscript, A.T. and D.M.S.; Funding acquisition, D.M.S., H.L., and J.C.M.G.; Supervision, D.M.S., H.L., and J.C.M.G.

## Competing interests

The authors declare no competing interests.
