## [Transparent Peer Review file · Nature Communications]

Mechanisms of gene regulation by SRCAP and H2A.Z

Corresponding Author: Professor David Suter

Version 0:

Reviewer comments:

Reviewer #1

(Remarks to the Author)

This manuscript presents a lot of data on SRCAP and H2A.Z. They use a degron line to disentangle the roles of SRCAP and H2A.Z. Although there appear to be interesting findings, it is not always clear how the authors arrive at their conclusions. Some of the presented data is inconsistent, and some conclusions are insubstantially supported by the data.

Comments:

- In the introduction, the authors state that 'mechanistic understanding of transcriptional regulation by SRCAP and H2A.Z remains obscured by conflicting evidence and technical challenges'. However, a clear statement, both in the abstract and in the introduction, clarifying how their approach is different compared to these previous studies, and how their approach circumvents these technical challenges, is missing. The most important difference in technical approach in this study is the use of rapid, inducible SRCAP degradation systems, yet this is not mentioned in the abstract, nor in the introduction.
- Figure 1A: Although this figure shows SRCAP-YA enrichment at H2A.Z peaks and active TSS, it does not show whether the enrichment is similar to WT-SRCAP, or not. In other words, is the enrichment of SRCAP-YA representative of WT-SRCAP enrichment at these peaks?
- Figure 1A/G: SRCAP-YA does not appear to be enriched at bivalent TSS peaks in figure 1A. Yet, H2A.Z appears the most enriched at bivalent TSS peaks in figure 1G. How can H2A.Z be so strongly enriched at bivalent TSS if SRCAP does not localize there?
- Figure 2A/2G: Why is the WT asynchronous panel similar to the WT mitotic panel in 2A, but a large difference presented in 2G?
- Figure 2B/S2B: Why are figures S2B and 2B different? In figure 2B, the H2A.Z enrichment increases at active enhancers, but in figure S2B, it decreases, the latter is also mentioned as a result in the text.
- Figure 2G/H: How can H2A.Z enrichment increase genome-wide (figure 2G) and specifically at bivalent promoters (figure 2H) in asynchronous cells when SRCAP is depleted? No potential explanation is given for this contradicting result. Why was the time-frame of only 1h IAA treatment chosen here? Figure 1 shows that H2A.Z enrichment is clearly decreased after 2h, but 1h was not tested.
- Very little detail was given about how promoter-enhancer loops are identified.
- Figure 4A: 'At 4 h of IAA, H3K4me3 levels were down in 66.6% of H3K4me3-enriched regions, with a marked decrease at bivalent promoters'. And: 'H2AK119ub1 displayed a massive increase at most enriched loci'. These sentences suggest that the authors only looked at the level of histone marks at loci that were enriched for that histone mark in the control cells. If you are interested in the effect of SRCAP/H2A.Z on these histone marks, why not look at loci that were enriched for SRCAP/H2A.Z in the control cells? After all, in the next part, to determine whether nucleosome properties explain the transcriptional changes, the authors did look at the H2A.Z enriched loci.
- Fig 4: Can the authors comment on why the amount of up/down regulated genes found by RNA seq are not reflected by the proteome results?
- "In contrast to SRCAP, H2A.Z was only slightly decreased in the whole chromatome (Fig.5f and Table S2), suggesting low or SRCAP-independent H2A.Z turnover in regions that are not recovered in ChIP-seq experiments, such as sonication-resistant heterochromatin 59." This is an interesting observation that the authors do not comment on. Why would this be specific to heterochromatin? This seems to relate to the steric hindrance model the authors propose.
- In the discussion the authors say "Several studies reported that the recruitment and nucleosome-remodeling activity of the BAF complex occur continuously on a time scale of minutes 33,85,86. Here we demonstrate dynamic SRCAP-mediated H2A.Z turnover over active and bivalent/poised regulatory elements on similar time scales." The authors measure of the course of several hours, which is not the same timescale as minutes, so this comparison is not valid.
- "We propose steric hindrance as a new mechanism by which bulky DNA-binding complexes can restrict the access of TFs

to chromatin." Although this seems a plausible explanation, steric hindrance by proteins as a mechanism for DNA accessibility is hardly revolutionary (the concept of heterochromatin), and should not be presented as a new mechanism. - 'We thus propose SRCAP/H2A.Z as a pivotal gene regulatory system operating in parallel to other epigenetic mechanisms to maintain cell identity and plasticity'. Given the minor changes in the proteome, this conclusion is not supported by the data. Especially since this is a near-complete knockdown of SRCAP, if it were a pivotal gene regulatory system, larger differences in the proteome would be expected.

Reviewer #2

(Remarks to the Author)

In this paper, Tollenaere et al demonstrated the mechanisms of gene regulation by SRCAP and H2A.Z in pluripotent stem cells. More importantly this study establishes the role of chromatin remodeler, which is independent of its catalytic activity, in comprehensively regulating the histone variants orchestrating the binding of transcription factors. This is an interesting finding in the field of cell fate determination for pluripotency by epigenetic regulation. The data are in general technically well performed. However, before publication, there are some points to be better clarified and corrected.

Major points

1. SRCAP-YA was rapidly degraded (Fig. 1b). It was mentioned that SRCAP was rapidly degraded, but the extent of degradation was not confirmed by data such as Western blot. Quantitative analysis of SRCAP protein levels over time (such as Western blot) could be added.
2. Fig. 1c showed the track of H2A.Z ChIP-seq. However, the track image of H2A ChIP-seq was absent, the author should add this track.
3. Loci with the highest H2A.Z turnover were initially highly enriched for SRCAP, the SWI/SNF, NuRD and Ep400 chromatin remodelers, and several PSC TFs (Fig. 1h and S1i), however, why the binding score of Ep400 is declined in the groups of decreased at 4h compared to the group of 2h, and then increased at 6h?
4. Although the article has analyzed different stages such as EG1, LG1, S, and G2/M, the detailed methods, strategies for cell cycle sorting, and possible cell cycle-dependent background factors have not been described adequately. Regarding cell cycle sorting, supplementary validation data from flow cytometry sorting should be provided in Fig. 2.
5. The protein levels of key upregulated genes (such as Gata3 and Nanog) have not been verified. The protein expression of key transcription factors can be confirmed by Western blotting in Fig. 3f.
6. Upregulated genes formed more promoter-enhancer loops than downregulated genes (Fig. 3e). which method? If used HiC, the author should calculate the E-P loop strength.
7. Using TOBIAS for TF footprint analysis provides important evidence for revealing the direct influence of SRCAP/H2A.Z on TF binding. However, the relevant parameters (such as significance threshold, pTF classification criteria, etc.) require more detailed explanations in terms of methodology.
8. This paper used MNase-seq to investigate the changes in nucleosome occupancy, protection fragment length, and "fragility". However, the results showed that even when H2A.Z was lost, there was no consistent change in the "fragility" of nucleosomes, indicating that this indicator may be more influenced by the genomic background. Further analyze the statistical significance of the structural changes of nucleosomes in different regulatory elements (such as active and bivalent promoters, and distal enhancers), and explore whether there are certain specific regions where the changes in TF binding are highly correlated. If possible, the in vitro experiments on nucleosome sliding or recombination to verify the catalytic independent function of SRCAP in H2A.Z deposition and nucleosome remodeling.
9. Has cdSRCAP completely lost its ability to deposit H2A.Z? If there is some residual activity, will it cause confusion in the downstream transcriptional effects? It is recommended to provide more quantitative data based on ChIP-seq or biochemical experiments to illustrate this point. For the various regulated genes classified, further discuss the biological significance of this differentiation mechanism in maintaining self-renewal and cell fate determination, as well as whether there are complementary or antagonistic effects.
10. This paper demonstrates the multi-level, both individual and collaborative, roles of SRCAP and H2A.Z in the cell cycle, TF binding, and transcriptional regulation. However, the discussion section has not yet established a systematic regulatory model. It is suggested to add a comprehensive schematic diagram to summarize the changes in H2A.Z deposition mediated by SRCAP, the modification of nucleosome state, the binding shielding by TF, and its impact on gene expression. This will help readers intuitively understand the interrelationships at various levels.
11. Considering that some changes in the TF footprints cannot be fully attributed to the direct action of SRCAP, it is suggested that the authors discuss and consider some potential alternative mechanisms (such as whether other chromatin remodeling factors are involved, or whether there are phenomena like RNA polymerase pausing, etc.), and list possible verification directions for the future in the discussion.

Minor points

1. Although statistical tests (such as pairwise Wilcoxon test) were conducted for all the data in this paper, but a part figure of the descriptions of key information such as the number of biological replicates, the sample size, and the data normalization method were not sufficient. The author should clearly specify the number of biological and technical replicates for each experiment, and provide more information on data distribution and sample size.
2. The font in the Fig 6i is too dense to be readable.
3. All the volcano charts lack the X-axis and Y-axis.

Version 1:

Reviewer comments:

Reviewer #1

(Remarks to the Author)

The authors have responded to our questions adequately. This has significantly improved the manuscript, and we now recommend it for publication.

Our only remaining confusion is the result in Figure 2 with the increase on bivalent promoters in asynchronous cells. Although we agree with the authors that the magnitude of this change is small, it is not that much smaller than the decrease for mitotic cells. In addition, it is significant, and we would have appreciated acknowledgement of this unexpected result in the manuscript.

Reviewer #2

(Remarks to the Author)

The author has addressed all my questions.

REVIEWER COMMENTS

Reviewer #1 (Remarks to the Author):

This manuscript presents a lot of data on SRCAP and H2A.Z. They use a degron line to disentangle the roles of SRCAP and H2A.Z. Although there appear to be interesting findings, it is not always clear how the authors arrive at their conclusions. Some of the presented data is inconsistent, and some conclusions are insubstantially supported by the data.

Comments:

1) In the introduction, the authors state that 'mechanistic understanding of transcriptional regulation by SRCAP and H2A.Z remains obscured by conflicting evidence and technical challenges'. However, a clear statement, both in the abstract and in the introduction, clarifying how their approach is different compared to these previous studies, and how their approach circumvents these technical challenges, is missing. The most important difference in technical approach in this study is the use of rapid, inducible SRCAP degradation systems, yet this is not mentioned in the abstract, nor in the introduction.

We thank the reviewer for this comment. We have now stated more clearly the strength of our study compared to previous studies in the abstract and the introduction.

2) Figure 1A: Although this figure shows SRCAP-YA enrichment at H2A.Z peaks and active TSS, it does not show whether the enrichment is similar to WT-SRCAP, or not. In other words, is the enrichment of SRCAP-YA representative of WT-SRCAP enrichment at these peaks?

We thank the reviewer for this point. We have now compared the enrichment of SRCAP in SYAT cells with CUT&RUN data in E14 wt mESCs from Patty et al. Cell Rep 2025 (PMID: 40450687) in revised Fig. S1e, cited on page 3 of the revised manuscript. By mapping the SRCAP signal in SYAT cells on peaks obtained in wt mESCs, we can conclude that SRCAP is enriched at the same locations in SYAT and in wt mESCs. Note that absolute score comparisons are not relevant here given that these experiments were performed in different labs.

3) Figure 1A/G: SRCAP-YA does not appear to be enriched at bivalent TSS peaks in figure 1A. Yet, H2A.Z appears the most enriched at bivalent TSS peaks in figure 1G. How can H2A.Z be so strongly enriched at bivalent TSS if SRCAP does not localize there?

We thank the reviewer for this important point. Indeed, we found very low SRCAP occupancy and high H2A.Z content at bivalent promoters, as well as a slower SRCAP-mediated H2A.Z turnover than at active promoters. The slow loss of H2A.Z upon SRCAP removal (as compared to active TSS) suggests that H2A.Z removal is slow in these regions. A slower eviction of H2A.Z is expected to lead to a slowed-down recruitment of SRCAP, as H2A.Z loading demands at these locations are lower than at active TSS. Therefore, **SRCAP enrichment is not expected to scale with H2A.Z occupancy, but with its rate of eviction**. The residual levels of H2A.Z after 8 hours of SRCAP removal at bivalent TSS could be explained either by their extremely slow turnover or the deposition of H2A.Z by other chromatin remodelers such as p400/Tip60. We have now explained this better on page 3 of the revised manuscript.

4) Figure 2A/2G: Why is the WT asynchronous panel similar to the WT mitotic panel in 2A, but a large difference presented in 2G?

We understand that comparing these two experiments can be confusing. This is explained by differences in the way these experiments were performed. In Fig. 2a we display results from ChIP-Seq performed on 2µg of chromatin from cells sorted for the different cell cycle phases or collected from an asynchronous sample. In contrast, in Fig. 2g, H2A.Z ChIP-Seq was performed on 650ng of chromatin from cells sorted according to Ypet levels (YPet-positive or negative) and cell cycle phase. Additionally, enrichment from experiments presented in Fig. 2g were mapped on peaks obtained from the experiment presented in Fig. 2a.

Below we represent the enrichment of the experiment presented in 2g but now mapped on peaks from that same experiment (respectively peaks obtained in mitosis or asynchronous sample). As expected, the enrichment is now similar in mitotic and asynchronous samples.

5) Figure 2B/S2B: Why are figures S2B and 2B different? In figure 2B, the H2A.Z enrichment increases at active enhancers, but in figure S2B, it decreases, the latter is also mentioned as a result in the text.

We thank the reviewer for raising this point. In Fig. 2b we represent H2A.Z enrichment across the cell cycle in promoters and enhancers, without removing regions where H2A.Z was not detected in order to include inactive regions where H2A.Z is typically absent or present at very low levels. In Fig. S2a and b, the violin plots display H2A.Z enrichment normalized on mean value in promoters and enhancers where at least one H2A.Z peak was detected. We apologize for the confusion and have now clarified this in the corresponding figure legends. Additionally, following the reviewer comment, we believe that the representations formerly presented in Fig. S2a and b are actually more informative, and are now presented as revised Fig. 2b, while former Fig. 2b is now revised Fig. S2b and c.

6) Figure 2G/H: How can H2A.Z enrichment increase genome-wide (figure 2G) and specifically at bivalent promoters (figure 2H) in asynchronous cells when SRCAP is depleted? No potential explanation is given for this contradicting result. Why was the time-frame of only 1h IAA treatment chosen here? Figure 1 shows that H2A.Z enrichment is clearly decreased after 2h, but 1h was not tested.

We thank the reviewer for pointing out this apparently paradoxical result and would like to provide some clarifications. We chose to treat cells for 1h as we wanted to assess the consequences of SRCAP loss at the G2/M transition, and because mitosis typically lasts around 45 minutes in mESCs. We thus decided to compare the changes in H2A.Z levels with asynchronous cells also treated with IAA for 1h. We did not further discuss the increase in H2A.Z levels after 1h of IAA treatment observed both genome-wide and at bivalent promoters because of its fold-change that is too small to be meaningful, even though we cannot fully exclude that such a mild increase is real and due to indirect effects such as increased H2A.Z deposition by the Ep400-Tip60 complex. However, because of the very small magnitude of this change, we prefer not to venture into discussing it in our manuscript.

7) Very little detail was given about how promoter-enhancer loops are identified.

We apologize for this oversight. We have now added more details in the Methods section (page 28 of the revised manuscript) regarding how loops were identified in the Enhancer Atlas database. Additionally, we have now used data from the Supplementary material of Hsieh et al., Molecular Cell 2020 (PMID: 32213323), listing all promoter-enhancer loops detected in their Micro-C data performed in wt mESCs, as well as the strength of these loops. We used these data to compute the number of loops formed by promoters of downregulated or upregulated genes as

well as the strength of these loops confirming the results obtained from Enhancer Atlas. These results are now presented as revised Fig. 3e, while the analysis initially presented in the main figure is now displayed in revised Fig. S3h.

8) Figure 4A: 'At 4 h of IAA, H3K4me3 levels were down in 66.6% of H3K4me3-enriched regions, with a marked decrease at bivalent promoters'. And: 'H2AK119ub1 displayed a massive increase at most enriched loci'. These sentences suggest that the authors only looked at the level of histone marks at loci that were enriched for that histone mark in the control cells. If you are interested in the effect of SRCAP/H2A.Z on these histone marks, why not look at loci that were enriched for SRCAP/H2A.Z in the control cells? After all, in the next part, to determine whether nucleosome properties explain the transcriptional changes, the authors did look at the H2A.Z enriched loci.

We understand the point made by the reviewer, and we have now included (revised Fig S4b, S4f and S4h) the same variation of H3K4me3, H3K27me3 and H2AK119ub1, respectively, in peaks shared with H2A.Z. These results are in line with those obtained when analyzing all peaks enriched for H3K4me3, H3K27me3 or H2AK119ub1.

9) Fig 4: Can the authors comment on why the amount of up/down regulated genes found by RNA seq are not reflected by the proteome results?

We apologize for not making this clear enough. Importantly, these experiments were performed after only 4 hours of IAA treatment. We see that at this time point, there are substantial changes in intronic RNA-Seq results, which reflect acute changes in transcriptional activity. We agree that these changes are expected to lead to corresponding changes in mature mRNAs and proteins, but this is expected to occur on much longer time scales than 4 hours. This is because mature mRNAs and proteins have half-lives on time scales of several hours to days. In line with this, after 4 hours of IAA treatment, we observe very limited changes in mature mRNAs levels (exonic reads), and thus the very minor changes in protein levels after 4 hours of IAA treatment are not surprising. We have now added a sentence on page 5 to clarify this.

10) "In contrast to SRCAP, H2A.Z was only slightly decreased in the whole chromatome (Fig.5f and Table S2), suggesting low or SRCAP- independent H2A.Z turnover in regions that are not recovered in ChIP-seq experiments, such as sonication-resistant heterochromatin 59." This is an interesting observation that the authors do not comment on. Why would this be specific to heterochromatin? This seems to relate to the steric hindrance model the authors propose.

We thank the reviewer for this comment, however we do not think that this observation relates to our steric hindrance model. As SRCAP is mostly enriched in accessible, active regions of the genome and nucleosome turnover is higher on these regions, we expect the maximum effect on H2A.Z levels on accessible regions that are also the easiest to map by ChIP-Seq. In contrast, closed, constitutive heterochromatin, especially at centromeres and telomeres that are also enriched for H2A.Z, is typically more resistant to chromatin fragmentation by sonication and composed of repeated sequences that are filtered out at the mapping state during ChIP-Seq reads processing. We now explain this better on page 7 of the revised manuscript.

11) In the discussion the authors say "Several studies reported that the recruitment and nucleosome-remodeling activity of the BAF complex occur continuously on a time scale of minutes 33,85,86. Here we demonstrate dynamic SRCAP-mediated H2A.Z turnover over active and bivalent/poised regulatory elements on similar time scales." The authors measure the course of several hours, which is not the same timescale as minutes, so this comparison is not valid.

We understand that this was interpreted as an overstatement, which we have now toned down (see revised manuscript, page 12). We however note that the large impact we see on H2A.Z levels as soon as SRCAP is fully degraded (after 2 hours of IAA) is compatible with SRCAP acting on a sub-hour time scale.

12) "We propose steric hindrance as a new mechanism by which bulky DNA-binding complexes can restrict the access of TFs to chromatin." Although this seems a plausible explanation, steric hindrance by proteins as a mechanism for DNA accessibility is hardly revolutionary (the concept of heterochromatin), and should not be presented as a new mechanism.

We apologize if our statement was interpreted as a novel concept in such a general manner. We do however believe that steric hindrance by a chromatin remodeler that hinders transcription factor binding is an unprecedented finding. We now better specify the nature of our novel findings on page 12 of the revised manuscript.

13) 'We thus propose SRCAP/H2A.Z as a pivotal gene regulatory system operating in parallel to other epigenetic mechanisms to maintain cell identity and plasticity'. Given the minor changes in the proteome, this conclusion is not supported by the data. Especially since this is a near-complete knockdown of SRCAP, if it were a pivotal gene regulatory system, larger differences in the proteome would be expected.

Here we disagree, since as we explain in the reply to comment 9, large changes in the proteome are not expected after only 4 hours of IAA, which we have now better emphasized on page 5 of the revised manuscript.

Reviewer #2 (Remarks to the Author):

In this paper, Tollenaere et al demonstrated the mechanisms of gene regulation by SRCAP and H2A.Z in pluripotent stem cells. More importantly this study establishes the role of chromatin remodeler, which is independent of its catalytic activity, in comprehensively regulating the histone variants orchestrating the binding of transcription factors. This is an interesting finding in the field of cell fate determination for pluripotency by epigenetic regulation. The data are in general technically well performed. However, before publication, there are some points to be better clarified and corrected.

Major points

1. SRCAP-YA was rapidly degraded (Fig. 1b). It was mentioned that SRCAP was rapidly degraded, but the extent of degradation was not confirmed by data such as Western blot. Quantitative analysis of SRCAP protein levels over time (such as Western blot) could be added.

SRCAP levels were both assessed by imaging of the Ypet fluorescent protein and mass spectrometry analysis of the proteome, confirming degradation of endogenous tagged SRCAP after 2 hours and 4 hours of IAA treatment, respectively. Following the reviewer comment and despite challenges in detecting the SRCAP protein because of its large molecular weight, we have now performed Western Blotting after different durations of IAA treatment for wt and SYAT mESCS treated with IAA for different time durations. We used both antibodies against SRCAP and GFP (which recognize the Ypet fluorescent protein fused to SRCAP). As expected, the Western Blots show a signal at 400kDa in SYAT in the control condition for both SRCAP and GFP, and these bands are not visible in the IAA-treated conditions. Results are displayed in revised Fig. S1f, and we added a corresponding Methods section on page 20 of the revised manuscript.

2. Fig. 1c showed the track of H2A.Z ChIP-seq. However, the track image of H2A ChIP-seq was absent, the author should added this track.

We thank the reviewer for this point. We initially did not include a track for H2A because in contrast to H2A.Z, its enrichment is very diffuse over the whole genome, and differences in overall enrichment are thus much easier to visualize in pile-up profiles. Nevertheless, we have now included genome tracks for H2A in the revised manuscript as Revised Fig. 1c.

3. Loci with the highest H2A.Z turnover were initially highly enriched for SRCAP, the SWI/SNF, NuRD and Ep400 chromatin remodelers, and several PSC TFs (Fig. 1h and S1i), however, why the binding score of Ep400 is declined in the groups of decreased at 4h compared to the group of 2h, and then increased at 6h?

We thank the reviewer for raising this interesting point. Loci that lose H2A.Z only after 6h of IAA treatment have a slower nucleosome turnover, which may favor the binding of Ep400. However, this statement is very speculative and we would prefer not including it in the final version of the manuscript.

4. Although the article has analyzed different stages such as EG1, LG1, S, and G2/M, the detailed methods, strategies for cell cycle sorting, and possible cell cycle-dependent background factors have not been described adequately. Regarding cell cycle sorting, supplementary validation data from flow cytometry sorting should be provided in Fig. 2.

We apologize for not describing these sorting strategies enough. Importantly, these were previously validated by several labs (Kadauke et al. 2012 (PMID: 22901805), Friman et al 2019 (PMID: 31794382), Placzek et al. 2025 (PMID: 40153434). We now have added FACS plots as revised Fig. S2a and in the Methods section (page 19 of the revised manuscript), additional information as well as citation of previous work from us and others that have used mitotic degrons and H3S10ph staining.

5. The protein levels of key upregulated genes (such as Gata3 and Nanog) have not been verified. The protein expression of key transcription factors can be confirmed by Western blotting in Fig.3f.

We would like to first state that in whole proteome analysis by Mass spectrometry, we didn't see any change in NANOG or GATA3 levels (Supplementary Table S2). Additionally, we have now added as revised Fig. S5h a Western Blot showing levels of NANOG in wt CGR8 as well as SYAT cells in control condition and treated with IAA for 2, 4, 6 and 8h, quantified on GAPDH levels, showing no change in total NANOG levels. We tried to analyze GATA3 levels the same way but failed at finding a proper antibody to recognize GATA3 in Western Blot. Also note that even after IAA treatment, we expect the levels of GATA3 to be very low as this transcription factor is typically expressed at later stages of differentiation, thus making it very difficult to blot in the context of pluripotent stem cells, while the high detection sensitivity of mass spectrometry analysis allowed us to detect GATA3.

6. Upregulated genes formed more promoter-enhancer loops than downregulated genes (Fig.3e). which method? If used HiC, the author should calculate the E-P loop strength.

Initially we had computed the number of loops formed by promoters of downregulated or upregulated genes using an existing database, Enhancer Atlas, which lists all detected enhancers in a given cell line and the promoter they interact with. This approach was used because genes relying more on enhancers for their regulation tend to form more diverse loops between their promoter and enhancers. We now added more information in the revised Methods section (page 28) to better explain this approach. Additionally, we have now used data from the Supplementary material of Hsieh et al. Mol Cell 2020 (PMID: 32213323), listing all promoter-enhancer loops detected in their Micro-C data performed in wt mESCs, as well as the strength of these loops. We used these data to compute the number of loops formed by promoters of downregulated or upregulated genes as well as the strength of these loops confirming the results obtained from Enhancer Atlas. These results are now presented as revised Fig. 3e while our initial analysis using the Enhancer Atlas is now displayed in revised Fig. S3h.

7. Using TOBIAS for TF footprint analysis provides important evidence for revealing the direct influence of SRCAP/H2A.Z on TF binding. However, the relevant parameters (such as significance threshold, pTF classification criteria, etc.) require more detailed explanations in terms of methodology.

We apologize for not justifying enough our choice for the significance threshold we used for TOBIAS analysis. We decided to place the threshold at $p < 0.0001$ and $|FC| > 0.05$ as with these parameters, changes in the negative control (wt CGR8 treated with IAA or vehicle) were very mild and not overlapping with the motifs that we found to be affected in SYAT cells treated with IAA. Our classification of TFs as pioneer TFs was based on two published lists (Lemma et al. 2022 (PMID: 35440061), Peng et al. 2024 (PMID: 38293962)), in which authors analyzed the ability of TFs to bind to chromatinized DNA and increase chromatin accessibility at bound regions in at least

some cell types. We have now added these explanations in the Methods section, page 30 of the revised manuscript.

8. This paper used MNase-seq to investigate the changes in nucleosome occupancy, protection fragment length, and "fragility". However, the results showed that even when H2A.Z was lost, there was no consistent change in the "fragility" of nucleosomes, indicating that this indicator may be more influenced by the genomic background. Further analyze the statistical significance of the structural changes of nucleosomes in different regulatory elements (such as active and bivalent promoters, and distal enhancers), and explore whether there are certain specific regions where the changes in TF binding are highly correlated.

We apologize that some statistical analyses were missing for the changes in nucleosome fragility, occupancy and fragment length distribution. We have now added statistical tests (Kruskal-Wallis rank sum test and pairwise Wilcoxon rank sum test) for differences in nucleosome occupancy and fragility in regions with different H2A.Z levels (Fig. 4f and h, Fig. S4k,m and n). Statistical analysis of the impact of IAA treatment on nucleosome occupancy and fragility was performed using Welch two samples t-test. Differences in nucleosome-protected fragment length distribution were statistically assessed using the Kolmogorov-Smirnov test. Results from all statistical tests performed in the study are now gathered in a supplementary table (Table S7).

We thank the reviewer for raising the potential link between nucleosome fragility and changes in TFs binding. We have now assessed nucleosome fragility at motifs predicted to gain or lose binding in TOBIAS after 4h of IAA treatment. Nucleosome fragility was not significantly changed after IAA treatment for motifs predicted to gain binding in TOBIAS. Because these results are not particularly informative, we decided not to include them in the revised manuscript.

If possible, the in vitro experiments on nucleosome sliding or recombination to verify the catalytic independent function of SRCAP in H2A.Z deposition and nucleosome remodeling.

We agree that this would be a very exciting experiment, however this would most likely require several years of work to purify and assemble the SRCAP complex in vitro and assay its catalytic-dependent in H2A.Z positioning and chromatin remodeling, and thus we think that this is out of scope of this manuscript.

9. Has cdSRCAP completely lost its ability to deposit H2A.Z? If there is some residual activity, will it cause confusion in the downstream transcriptional effects? It is recommended to provide more quantitative data based on ChIP-seq or biochemical experiments to illustrate this point. For the various regulated genes classified, further discuss the biological significance of this differentiation mechanism in maintaining self-renewal and cell fate determination, as well as whether there are complementary or antagonistic effects.

cdSRCAP is an ATPase dead mutant of SRCAP and as the SRCAP ATPase activity was shown to be necessary for H2A-H2B dimer eviction and H2A.Z-H2B deposition (Girvan et al. Nature 2024, PMID: 39506114), cdSRCAP is expected to be defective for H2A.Z deposition. However, to satisfy the reviewer request, we have now compared H2A.Z genomic enrichment by ChIP-Seq between cells overexpressing cdSRCAP or not (+/- dox in the cdSRCAP inducible SYAT cell line), and in the presence or absence of endogenous SRCAP (+/- IAA). We

found that cells overexpressing cdSRCAP have a slightly lower level of H2A.Z in regions where H2A.Z is present, indicating that cdSRCAP does not deposit H2A.Z and even plays a slight dominant negative role on H2A.Z deposition. These data are now presented as revised Fig. S6c and mentioned on page 9 of the revised manuscript.

Among the six RNA clusters identified, two are independently regulated by SRCAP or H2A.Z. For these clusters (A and E), the depletion of H2A.Z or SRCAP will respectively lead to upregulation of genes from cluster A and downregulation of genes from cluster E, as we observed upon IAA treatment. For all four other clusters, H2A.Z and SRCAP have antagonistic regulatory effects on transcription. For some genes, H2A.Z and SRCAP compensate for each other activity (cluster C) as confirmed by the absence of transcriptional changes upon H2A.Z/SRCAP depletion. We expect the expression of these genes to be buffered by H2A.Z/SRCAP in mESCs, favoring cell identity maintenance. For some genes (cluster B), the repressive impact of H2A.Z dominates on the activatory impact of SRCAP. We observe the opposite for genes in cluster D. SRCAP deposits H2A.Z in H2A-containing nucleosomes, its binding being inhibited by enrichment of H2A.Z. We believe that this feedback loop could create windows of opportunities upon signalling cues for upregulation or downregulation of these genes. We now discuss these results in the discussion section on pages 12-13 of the revised manuscript.

10. This paper demonstrates the multi-level, both individual and collaborative, roles of SRCAP and H2A.Z in the cell cycle, TF binding, and transcriptional regulation. However, the discussion section has not yet established a systematic regulatory model. It is suggested to add a comprehensive schematic diagram to summarize the changes in H2A.Z deposition mediated by SRCAP, the modification of nucleosome state, the binding shielding by TF, and its impact on gene expression. This will help readers intuitively understand the interrelationships at various levels.

We thank the reviewer for this very good suggestion. We have now added a schematic diagram as revised Fig. 8 to summarize our findings on H2A.Z turnover and how SRCAP loss affects nucleosome properties, TF binding and transcription.

11. Considering that some changes in the TF footprints cannot be fully attributed to the direct action of SRCAP, it is suggested that the authors discuss and consider some potential alternative mechanisms (such as whether other chromatin remodeling factors are involved, or whether there are phenomena like RNA polymerase pausing, etc.), and list possible verification directions for the future in the discussion.

We thank the reviewer for their suggestion, we have now included additional discussion points on page 12 of the revised manuscript.

Minor points

1. Although statistical tests (such as pairwise Wilcoxon test) were conducted for all the data in this paper, but a part figure of the descriptions of key information such as the number of biological replicates, the sample size, and the data normalization method were not sufficient. The author should clearly specify the number of biological and technical replicates for each experiment, and provide more information on data distribution and sample size.

For heatmaps of tag counts and violin plots, we have now added sample sizes corresponding to the number of studied peaks or regulatory regions directly on the plots or in the legend of the figure (in Fig. 1, S1, 2, S2, 4 and S4, 5, 6 and S6). For the number of nucleosomes quantified in the different regulatory regions and depending on H2A.Z levels, we have appended a new table containing this information to the supplementary material (Table S6). Missing numbers of biological or technical replicates were added in the Methods section at the beginning of the paragraph describing the experimental procedure for each given type of experiment.

2. The font in the Fig 6i is too dense to be readable.

The font of Fig. 6i has now been enlarged.

3. All the volcano charts lack the X-axis and Y-axis.

We understand the point, but we believe that keeping the volcano plots without axes that are drawn as lines (noting that the legends and graduations are present) is better for readability.

Response to reviewers comments

Reviewer #1 (Remarks to the Author):

The authors have responded to our questions adequately. This has significantly improved the manuscript, and we now recommend it for publication.

Our only remaining confusion is the result in Figure 2 with the increase on bivalent promoters in asynchronous cells. Although we agree with the authors that the magnitude of this change is small, it is not that much smaller than the decrease for mitotic cells. In addition, it is significant, and we would have appreciated acknowledgement of this unexpected result in the manuscript.

We have now added the following sentence on page 4 of the manuscript "While we noted a small increase in H2A.Z enrichment at bivalent promoters in asynchronous cells upon 1h IAA treatment, the biological meaning of this finding is unclear (Fig.2h)."

Reviewer #2 (Remarks to the Author):

The author has addressed all my questions.